# Using the president's tweets to understand political diversion in the age of social media

Stephan Lewandowsky ⓘ [1,2 ✉], Michael Jetter[2,3,4] & Ullrich K. H. Ecker[2]

Social media has arguably shifted political agenda-setting power away from mainstream media onto politicians. Current U.S. President Trump's reliance on Twitter is unprecedented, but the underlying implications for agenda setting are poorly understood. Using the president as a case study, we present evidence suggesting that President Trump's use of Twitter diverts crucial media (The New York Times and ABC News) from topics that are potentially harmful to him. We find that increased media coverage of the Mueller investigation is immediately followed by Trump tweeting increasingly about unrelated issues. This increased activity, in turn, is followed by a reduction in coverage of the Mueller investigation—a finding that is consistent with the hypothesis that President Trump's tweets may also successfully divert the media from topics that he considers threatening. The pattern is absent in placebo analyses involving Brexit coverage and several other topics that do not present a political risk to the president. Our results are robust to the inclusion of numerous control variables and examination of several alternative explanations, although the generality of the successful diversion must be established by further investigation.

[1] University of Bristol, 12A Priory Road, Bristol BS8 1TU, UK. [2] University of Western Australia, 35 Stirling Hwy, Perth, WA 6009, Australia. [3] IZA, Schaumburg-Lippe-Straße 5-9, Bonn 53113, Germany. [4] CESifo, Poschingerstr. 5, Munich 81679, Germany. ✉email: stephan.lewandowsky@bristol.ac.uk

On August 4, 2014, a devastating earthquake maimed and killed thousands in China's Yunnan province. Within hours, Chinese media were saturated with stories about the apparent confession by an Internet celebrity to have engaged in gambling and prostitution. News about the earthquake was marginalized, to the point that the Chinese Red Cross implored the public to ignore the celebrity scandal. The flooding of the media with stories about a minor scandal appeared to have been no accident, but represented a concerted effort of the Chinese government to distract the public's attention from the earthquake and the government's inadequate disaster preparedness[1]. This organized distraction was not an isolated incident. It has been estimated that the Chinese government posts around 450 million social media comments per year[2], using a 50-cent army of operatives to disseminate messages. Unlike traditional censorship of print or broadcast media, which interfered with writers and speakers to control the source of information, this new form of Internet-based censorship interferes with consumers by diverting attention from controversial issues. Inconvenient speech is drowned out rather than being banned outright[3].

In Western democracies, by contrast, politicians cannot orchestrate coverage in social and conventional media to their liking. The power of democratic politicians to set the political agenda is therefore limited, and it is conventionally assumed that it is primarily the media, not politicians, that determine the agenda of public discourse in liberal democracies[4,5]. Several lines of evidence support this assumption. For example, coverage of terrorist attacks in the New York Times has been causally linked to further terrorist attacks, with one additional article producing 1.4 attacks over the following week[6]. Coverage of al-Qaeda in premier US broadcast and print media has been causally linked to additional terrorist attacks and increased popularity of the al-Qaeda terrorist network[7]. Similarly, media coverage has been identified as a driver—rather than an echo—of public support for right-wing populist parties in the UK[8]. Further support for the power of the media emerges from two quasi-experimental field studies in the UK. In one case, when the Sun tabloid switched its explicit endorsement from the Conservative party to Labour in 1997, and back again to the Conservatives in 2010, each switch translated into an estimated additional 500,000 votes for the favored party at the next election[9]. In another case, the long-standing boycott of the anti-European Sun tabloid in the city of Liverpool (arising from its untruthful coverage of a tragic stadium incident in 1989 with multiple fatalities among Liverpool soccer fans), rendered attitudes towards the European Union in Liverpool more positive than in comparable areas that did not boycott the Sun[10]. In the United States, the gradual introduction of Fox News coverage in communities around the country has been directly linked to an increase in Republican vote share[11]. Finally, a randomized field experiment in the US that controlled media coverage of local papers by syndicating selected topics on randomly chosen days, identified strong flow-on effects into public discourse. The intervention increased public discussion of an issue by more than 60%[4].

This conventional view is, however, under scrutiny. More nuanced recent analyses have invoked a market in which the elites, mass media, and citizens seek to establish an equilibrium[12]. In particular, the rapid rise of social media, including the microblogging platform Twitter, has provided new avenues for political agenda setting that have increasingly discernible impact. For example, the content of Twitter discussions of the HPV vaccine explains differences in vaccine uptake beyond those explainable by other socioeconomic variables. Greater spread of misinformation and conspiracy theories on Twitter are associated with lower vaccination rates[13]. Similarly, fake news (fabricated stories that are presented as news on social media) have

controlled the popularity of many issues in US politics[14], mainly owing to the responsiveness of partisan media outlets. The entanglement of partisan media and social media is of considerable generality and can sometimes override the agenda-setting power of leading outlets such as the New York Times[15].

One important characteristic of Twitter is that it allows politicians to directly influence the public's political agenda[16]. For example, as early as 2012, a sample of journalists acknowledged their reliance on Twitter to generate stories and obtain quotes from politicians[17]. With the appearance of Donald Trump on the political scene, Twitter has been elevated to a central role in global politics. President Trump has posted around 49,000 tweets as of February 2020. To date, research has focused primarily on the content of Donald Trump's tweets[18–21]. Relatively less attention has been devoted to the agenda-setting role of the tweets. Some research has identified the number of retweets Trump receives as a frequent positive predictor of news stories[22] (Though see[16] for a somewhat contrary position.). During the 2016 election campaign, Trump's tweets on average received three times as much attention than those of his opponent, Hillary Clinton, suggesting that he was more successful at commanding public attention[23].

Here we focus on one aspect of agenda-setting, namely Donald Trump's presumed strategic deployment of tweets to divert attention away from issues that are potentially threatening or harmful to the president[24,25]. Unlike the Chinese government, which has a 50-cent army at its disposal, diversion can only work for President Trump if he can directly move the media's or the public's attention away from harmful issues. Anecdotally, there are instances in which this diversion appears to have been successful. For example, in late 2016, President-elect Trump repeatedly criticized the cast of a Broadway play via Twitter after the actors publically pleaded for a "diverse America." This Twitter event coincided with the revelation that Trump had agreed to a $25 million settlement (including a $1 million penalty[26]) of lawsuits against his (now defunct) Trump University. An analysis of people's internet search behavior using Google Trends confirmed that the public showed far greater interest in the Broadway controversy than the Trump University settlement[27], attesting to the success of the presumed diversion. However, to date evidence for diversion has remained anecdotal.

This article provides the first empirical test of the hypothesis that President Trump's use of Twitter diverts attention from news that is politically harmful to him. In particular, we posit that any increase in harmful media coverage may be followed by increased diversionary Twitter activity. In turn, if this diversion is successful, it should depress subsequent media coverage of the harmful topic. To operationalize the analysis, we focused on the Mueller investigation as a source of potentially threatening or harmful media coverage. Special prosecutor Robert Mueller was appointed in March 2017 to investigate Russian interference in the 2016 election and potential links between the Trump campaign and Russian officials. Given that legal scholars discussed processes by which a sitting president could be indicted even before Mueller delivered his report[28,29], and given that Mueller indicted associates of Trump during the investigation[30], there can be no doubt that this investigation posed a serious political risk to Donald Trump during the first 2 years of his presidency.

The center panel of Fig. 1 provides an overview of the presumed statistical paths of this diversion. Any increase in harmful media coverage, represented by the word cloud on the left, should be followed by increased diversionary Twitter activity, which is captured by the word cloud on the right. The expected increase is represented by the "+" sign along that path. If this diversion is successful, it should depress subsequent media coverage of the harmful topic (path labeled by "−" to represent the opposite direction of the association). Each of the two paths is represented

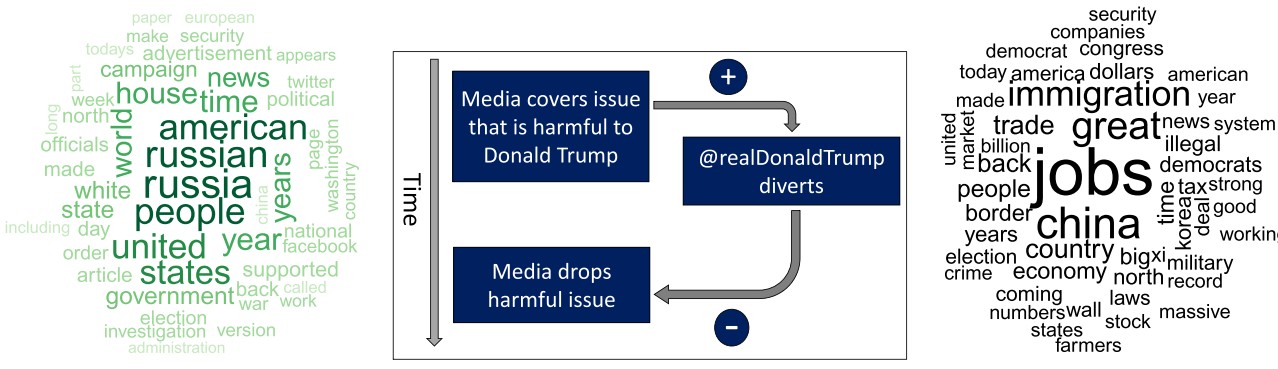

**Fig. 1 The center panel shows a conceptual model of potential strategic diversion by Donald Trump via Twitter (where his handle is @realDonaldTrump).** The word cloud on the left contains the 50 most frequent words from all articles in the NYT that contained "Russia" or "Mueller" as keywords. The NYT articles captured by the word cloud contained a total of 146,307 unique words. We excluded "president" and "Trump" because of their outlying high frequency. In addition to the 50 words shown here, the top 1% of high-frequency items included terms such as "collusion," "impeachment," "conspiracy," and numerous names of actors relevant to the investigation, such as "Mueller," "Putin," "Comey," "Manafort," and so on. The word cloud on the right represents the 50 most frequent words occurring in Donald Trump's tweets that were chosen, on the basis of keywords, to represent his preferred topics. The expected sign of the regression coefficient is shown next to each path and is identical for both approaches (OLS and 3SLS). The delay between media coverage and diversionary tweets is assumed to be shorter than the subsequent response of the media to the diversion. This reflects the relative sluggishness of the news cycle compared to Donald Trump's instant-response capability on Twitter.

by a regression model that relates Twitter activity (represented by the number of relevant tweets) with media coverage (represented by the number of reports concerning Russia-Mueller). We approached the analysis in two ways: first, we used ordinary least squares (OLS) in which each of the two paths was captured by a separate regression model. Second, we used three-stage least squares (3SLS) to estimate both path coefficients in a single model simultaneously.

We obtained daily news coverage from two acknowledged benchmark sources in TV and print media during the first 2 years of Trump's presidency (20 January 2017 – 20 January 2019): The American Broadcasting Corporation's *ABC World News Tonight*, a 30-min daily evening news show that ranks first among all evening news programs in the US[31], and the *New York Times* (NYT), which is widely considered to be the world's most influential newspaper by leaders in business, politics, science, and culture[32]. The NYT was also ranked first in the world, based on traffic flow, by the international search engine 4IMN in 2016 (https://www.4imn.com/top200/). A recent quantitative analysis of more than 800,000 news articles in the NYT through a combination of machine learning and human judgment (involving a sample of nearly 800 judges) has identified the *New York Times* as being quite close to the political center, with a slight liberal leaning (which was found to be smaller than, e.g., the corresponding conservative slant of *The Wall Street Journal*)[33]. Similarly, ABC News is known to be favored by centrist voters without however being shunned by partisans on either side[34]. In the online news ecosystem, ABC (combined with Yahoo!) and NYT form some of the most central nodes in the network, and nearly the entire online news audience tends to congregate at those brand-name sites[35].

Specification of diversionary topics on Twitter is challenging a priori because potentially any topic, other than the Mueller investigation itself, could be recruited by the president as a diversionary effort. We addressed this problem in two ways. First, we conducted a targeted analysis in which the diversionary topics were stipulated a priori to be those that President Trump prefers to talk about, based on our analysis of his political position and rhetoric during the first 2 years of this presidency. Second, we conducted an expanded analysis that considered the president's entire Twitter vocabulary as a potential source of diversion. This analysis allowed for the possibility that Trump would

divert by highlighting topics other than those that he consistently favors.

Both analyses included a number of controls and robustness checks, such as randomization, sensitivity analyses, and the use of placebo keywords, to rule out artifactual explanations. Both analyses were approached in two different ways. The first approach fitted two independent ordinary least squares (OLS) regression models that (a) predicted diversion from Mueller coverage and (b) captured a suppression of Mueller coverage in the media as a downstream consequence of diversion (see Fig. 1 and "Methods"). The second approach used a three-stage least squares (3SLS) regression model. In a 3SLS regression, multiple equations are estimated simultaneously; in our case there are two equations that capture diversion and suppression, respectively. This approach is particularly suitable when phenomena may be reciprocally causal.

## Results

**Targeted analysis**. The targeted analysis focused on the association between media coverage of the Mueller investigation and President Trump's use of Twitter to divert attention from that coverage. The analysis also asked whether that diversion, if it is triggered, might in turn suppress media coverage of the Mueller investigation.

We assumed that the president's tweets would divert attention from Mueller to his preferred topics. We considered the three keywords "China," "jobs," and "immigration" as markers of those topics and explored all combinations of those words being used in Trump's tweets. Our choice of keywords was based on the following considerations. First, at the time of analysis, which predates the COVID-19 crisis, the US unemployment rate was at its lowest in at least a decade (3.8% in 2019, monthly rates provided by the Bureau of Labour Statistics averaged through June; https://www.bls.gov/webapps/legacy/cpsatab1.htm), and President Trump is routinely claiming credit for job creation[36]. The president has also made China-related issues some of his main international policy topics (e.g., when it comes to international trade)[37], suggesting that this is also one of his focal areas of political activity. Finally, controlling and curtailing immigration was central to Trump's election campaign and continues to be a major policy plank. In further support of our choice of keywords, an analysis of Donald Trump's campaign

**Table 1 Predicting diversionary tweets (CJI) from threatening media coverage (Russia-Mueller; columns 1–3) and predicting suppression from diversionary tweets (columns 4–6).**

| Dependent variable | CJI tweets$_t$ | | | NYT[a] | ABC[a] | Average[b] |
|---|---|---|---|---|---|---|
| | (1) | (2) | (3) | (4) | (5) | (6) |
| NYT Russia/Mueller$_t$ | 0.012 (0.004) $p = 0.005$ [0.005] $p = 0.007$ | | | | | |
| ABC Russia/Mueller$_t$ | | 0.213 (0.084) $p = 0.012$ [0.078] $p = 0.006$ | | | | |
| Average Russia/Mueller$_t$ | | | 0.226 (0.067) $p = 0.001$ [0.062] $p = 0.000$ | | | |
| CJI tweets$_{t-1}$ | ° | ° | ° | −0.403 (0.207) $p = 0.052$ [0.219] $p = 0.066$ | −0.038 (0.020) $p = 0.052$ [0.020] $p = 0.061$ | −0.050 (0.020) $p = 0.016$ [0.022] $p = 0.024$ |
| N | 721 | 721 | 721 | 703 | 710 | 703 |

Each model included control variables for each week during the sampling period and long-term time trends as well as the appropriate number of lagged observations for the dependent variable (see "Methods"). Table entries are coefficients (OLS standard errors) [Newey-West standard errors].
[a]Coverage of Russia-Mueller on day $t$.
[b]Average of the standardized values of coverage of NYT and ABC.
°Lagged predictor is shown only when of interest.

promises in 2016 by independent fact-checker *Politifact* revealed that the top 5 promises were consonant with our topic keywords[38]. Early in his term, and during our sampling period, job growth in the US and withdrawing from trade agreements together with action on immigration were again identified as being among the three issues the president "got most right"[39]. At the time of this writing, a website by the Trump campaign (https://www.promiseskept.com/) that is recording the president's accomplishments lists "Economy and Jobs," "Immigration," and "Foreign Policy" as the first three items. The "Foreign Policy" page, in turn, mentions China 12 times (with another nine occurrences of "Chinese"). The only other countries mentioned more than once were Israel (9), Iran (4), Canada (3), and Japan (3). The word cloud on the right of Fig. 1 summarizes the content of the diversionary tweets by showing the 50 most frequent words used in the tweets (omitting function words and stop words).

Table 1 summarizes the results for two different variants of a pair of independent OLS linear regression models using those keywords (see "Methods" for details). Standard errors derived from conventional OLS analyses are displayed in parentheses, whereas Newey-West adjusted standard errors that accommodate potential autocorrelations are reported in brackets (see "Methods" for a detailed discussion of the variables in the regression models). The Supplementary information reports a full exploration of the autocorrelation structure of the data (Tables S2–S5). All analyses included the relevant lags to model autocorrelations. The first model predicted diversion, represented by the number of times the three diversionary keywords appeared in tweets, from adverse media coverage on the same day and is shown in the first three columns of the table. If the president's tweets about his preferred issues lead to diversion, then regression coefficients for the diversion model should be positive and statistically significant. The table shows that this was indeed observed for all media coverage (NYT, ABC, and the combination of the two formed by averaging their standardized coverage). The magnitude of the associations is illustrated in the top row of panels in Fig. 2. Numerically, each additional *ABC News* headline containing Russia or Mueller would have been associated with 0.2 additional mentions of one of our keywords in tweets (column 2 in Table 1).

The second model, which predicted media coverage as a function of the number of diversionary tweets on the previous day, is shown in the rightmost three columns in Table 1. If the diversion was successful, then these regression coefficients are expected to be negative, indicating suppression of potentially

harmful media coverage by the president's tweets. The table shows that threatening media coverage was negatively associated with diversionary tweets. The magnitude of the association is illustrated in the bottom row of panels in Fig. 2. Each additional keyword mention in tweets is associated with a decrease of nearly one-half of an occurrence of "Russia" or "Mueller" from the next day's NYT (column 4 in Table 1). Table 1 provides an existence proof for the relationships of interest. The Supplementary information reports an additional, more nuanced set of analyses for different combinations and subsets of the three critical keywords (Tables S7–S12). These analyses generally confirm the overall pattern in Table 1.

One problematic aspect of this initial analysis is that artifactual explanations for the pattern cannot be ruled out. In particular, although one interpretation of these estimates is consistent with our hypotheses of (1) media coverage causing diversion and (2) the diversion in turn suppressing media coverage, the available data do not permit an unequivocal interpretation. Specifically, remaining endogeneity concerns (measurement error, reverse causality, and omitted variables) threaten a pure interpretation of these results as causal. We address each of those concerns in turn, and the associated conclusions suggest endogeneity would be unlikely to fully explain away our findings.

First, measurement error is unlikely to explain the relationships we found given that we draw from the universe of all NYT articles, ABC News segments, and Trump tweets in our sample period. Even if we were to miss some articles, news segments, or tweets (perhaps because our keywords did not fully catch all relevant articles or news segments), it is not clear how this would produce a systematic bias in one direction that could fundamentally influence our estimates. We support this judgment by displaying word clouds of all selected content (e.g., Fig. 1). If we systematically mismeasured the content of tweets and media coverage, the word clouds would reveal the error through intrusion of unexpected content or absence of content that would be expected to be present based on knowledge of the topic.

Similarly, there are reasons to believe that reverse causality cannot fully account for our results. For the first model (diversion; columns 1-3 in Table 1), it is less likely that Trump's diversionary tweets could generate more news about the Mueller investigation on the same day than the reverse, namely that more news generate more tweets. Given the lag time of news reports, even in the digital age, there is limited opportunity for a tweet to generate coverage within the same 24-h period. Moreover, even

## Association between media coverage and diversionary tweets

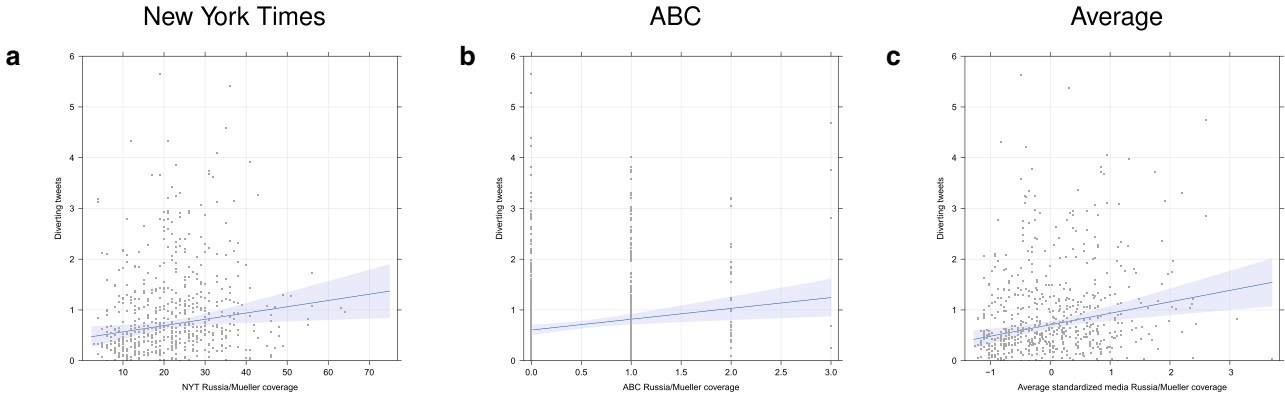

## Association between diversionary tweets and media coverage

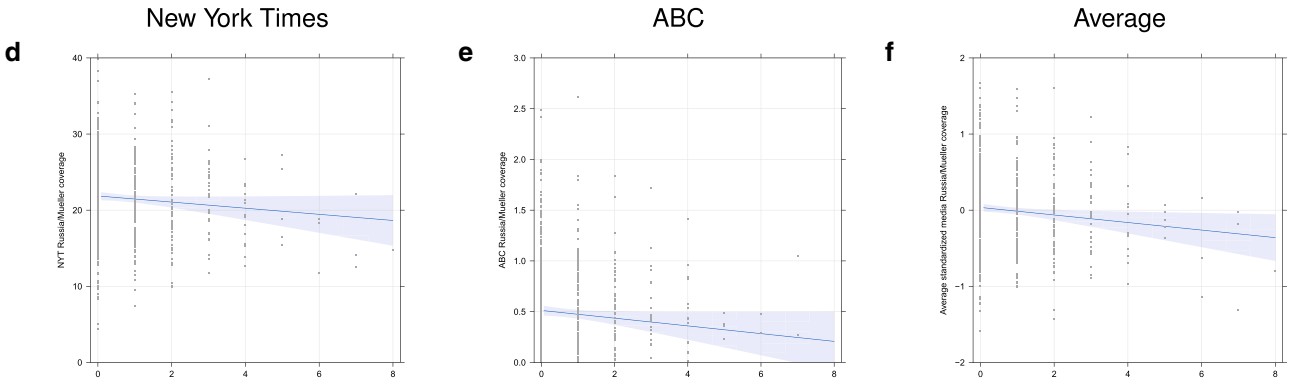

**Fig. 2 Association between media coverage of Russia/Mueller and diversionary tweets (top row of panel), and association between diversionary tweets and subsequent media coverage (bottom row of panels).** NYT is shown in panels (**a**, **d**); ABC news in panels (**b**, **e**); and the average of the two in panels (**c**, **f**).

putting aside that timing constraint, there is no obvious mechanism that would explain why the media systematically reports more on Mueller/Russia because President Trump tweeted on "China," "jobs," or "immigration"—we are not able to formulate a hypothesis why the media would respond in this manner. The reverse, however, motivated our analysis, namely the hypothesis that when Trump is confronted with uncomfortable media coverage, he tweets about unrelated topics. For the suppression model (columns 4–6 in Table 1), the fact that we predict news today with Trump's tweets from yesterday reduces concerns about reverse causality. One might still entertain the possibility that Trump's tweets divert pre-emptively, in expectation of Mueller coverage tomorrow. However, that would, if anything, work against us detecting suppression. If Trump's tweets are pre-emptive, we should see more tweets on days prior to increased Mueller coverage, thus creating a positive association. However, we observed a statistically significant negative relationship, which is the opposite of what is expected under the hypothetical anticipation scenario. The negative association is, however, entirely consonant with our hypothesis that diversion may be effective and may reduce inconvenient media coverage.

By contrast, we consider the hidden role of omitted variables to be the largest threat to causal identification. In the absence of controlled experimentation (or another empirical identification strategy suited to identify causality), one can never be certain that an effect is not caused by hidden omitted variables that interfere in the presumed causal path. This is an in-principle problem that

no observational study can overcome with absolute certainty. It is, however, possible to test whether omitted variables are likely to explain the observed pattern. Our first line of attack was to conduct a sensitivity analysis to obtain a robustness value for the diversion and suppression models involving average media coverage (columns 3 and 6 in Table 1)[40]. The robustness value captures the minimum strength of association that any unobserved omitted variables must have with the variables in the model (predictor and outcome) to change the statistical conclusions. The details of the sensitivity analysis are reported in the Supplementary information (Fig. S1 and Table S6). The results further lend support to our hypothesis that adverse media coverage causes the president to engage in diversion, and that this diversion, in turn, causes the media to reduce that coverage, although endogeneity from potentially omitted variables remains less likely to be a concern for the diversion model than the suppression model (see Fig. S1 and Table S6 for detailed quantification).

We additionally tackled the omitted-variable problem by fitting both models (diversion and suppression) simultaneously using 3SLS (see "Methods"). Table 2 reveals that the 3SLS results replicated the overall pattern of the OLS analysis, although the significance of the suppression is attenuated. A noteworthy aspect of our 3SLS analysis is that it used two ways to model the temporal offset between tweets and subsequent, potentially suppressed, media coverage. In panel A in Table 2, we used yesterday's tweets to predict today's coverage. This parallels the

**Table 2 Three-stage least squares (3SLS) models to predict diversionary tweets (CJI) from threatening media coverage (Russia-Mueller; columns 1–3) and predicting suppression from diversionary tweets (columns 4–6) simultaneously.**

| Dependent variable | CJI tweets$_t$ | | | NYT[a] | ABC[a] | Average[b] |
|---|---|---|---|---|---|---|
| | (1) | (2) | (3) | (4) | (5) | (6) |
| Panel A: yesterday's tweets predict today's coverage[c] | | | | | | |
| NYT Russia/Mueller$_t$ | 0.018 (0.006) $p = 0.007$ | | | | | |
| ABC Russia/Mueller$_t$ | | 0.225 (0.190) $p = 0.236$ | | | | |
| Average Russia/Mueller$_t$ | | | 0.293 (0.111) $p = 0.009$ | | | |
| CJI tweets$_{t-1}$ | ° | ° | ° | −0.399 (0.207) $p = 0.054$ | −0.038 (0.018) $p = 0.037$ | −0.050 (0.019) $p = 0.010$ |
| N | 703 | 710 | 703 | 703 | 710 | 703 |
| Panel B: today's tweets predict tomorrow's coverage[c] | | | | | | |
| NYT Russia/Mueller$_t$ | 0.012 (0.005) $p = 0.009$ | | | | | |
| ABC Russia/Mueller$_t$ | | 0.220 (0.068) $p = 0.001$ | | | | |
| Average Russia/Mueller$_t$ | | | 0.228 (0.060) $p = 0.000$ | | | |
| CJI tweets$_t$ | ° | ° | ° | −0.732 (0.620) $p = 0.238$ | −0.090 (0.054) $p = 0.095$ | −0.094 (0.057) $p = 0.099$ |
| N | 702 | 709 | 702 | 702 | 709 | 702 |

[a]Coverage of Russia-Mueller on day $t$ (panel A) or day $t+1$ (panel B).
[b]Average of the standardized values of coverage of NYT and ABC.
[c]Each model included control variables for each week during the sampling period and long-term time trends as well as the appropriate number of lagged observations for the dependent variable (see "Methods"). Table entries are coefficients (standard errors).
°Lagged predictor is shown only when of interest.

OLS analyses from Table 1. In panel B, by contrast, we predicted tomorrow's coverage from today's tweets. The pattern is remarkably similar across both panels: There are strong and nearly-uniformly statistically significant coefficients of adverse media coverage predicting diversion. Conversely, all coefficients for suppression are negative, although their level of significance is more heterogeneous.

The 3SLS results further diminish the likelihood of an artifactual explanation: for omitted variables to explain the observed joint pattern of diversion and suppression, those confounders would have to simultaneously explain a positive association between two variables on the same day and a negative association from one day to the next across two different intervals —namely from yesterday to today as well as from today to tomorrow. Moreover, those omitted variables would have to exert their effect in the presence of more than 100 other control variables and a large number of lagged variables. We consider this possibility to be unlikely.

Finally, to further explore whether the observed pattern of diversion and suppression was a specific response to harmful coverage, we conducted a parallel analysis using Brexit as a placebo topic. Like the Mueller inquiry, Brexit was a prominent issue throughout most of the sampling period and not under President Trump's control. Unlike Mueller, however, Brexit was not potentially harmful to the president—on the contrary, British campaigners to leave the European Union were linked to Trump and his team[41]. Table 3 shows the results of a model predicting diversionary tweets using the same three Twitter keywords but NYT coverage of Brexit as a predictor (using 24 days of lagged variables as suggested by an analysis of autocorrelations). Figure 3 illustrates the content of the Brexit coverage and confirms that the topic does not touch on issues that are likely to be politically harmful to the president. ABC News did not report on Brexit with sufficient frequency to permit analysis. Neither of the coefficients involving NYT are statistically significant, as one would expect for media coverage that is of no concern to the president.

To provide a more formal contrast between Brexit and Russia-Mueller, we combined the two models (Brexit: column 1 of Table 3; Russia-Mueller: column 1 of Table 1) into a single system of equations for a Seemingly Unrelated Regression (SUR) analysis[42]. Within a SUR framework, the consequences of constraining individual parameters can be jointly estimated for the two models. We found that forcing the coefficient for NYT Russia-Mueller coverage to be zero led to a significant loss of fit, $\chi^2(1) = 6.15$, $p = 0.013$, whereas setting the coefficient to zero for Brexit coverage had no effect, $\chi^2(1) = 1.41$, $p > 0.1$. (Forcing both coefficients to be equal entailed no significant loss of fit, owing to the imprecision with which the Brexit coefficient was estimated, with a 95% confidence interval that spanned zero and was nearly five times wider than for Russia-Mueller.)

Considered as a whole, these targeted analyses suggest that, during the sampling period, President Trump's tweets about his preferred topics diverted attention from inconvenient media coverage. That diversion, in turn, appears to be followed by suppression of the inconvenient coverage. Because we have no experimental control over the data, this conclusion must be caveated by allowing for the possibility that the results instead reflect the operation of hidden variables. However, additional analyses to explore that possibility produce results to discount that possibility, at least for the diversion model. We acknowledge that the status of the suppression effect is less robust in statistical terms. The expanded analysis further buttresses our conclusion by showing its generality and robustness.

**Expanded analysis**. Although the Twitter keywords for the targeted analysis were chosen to reflect the president's preferred topics, the president may divert using other issues as well. The expanded analysis therefore considered all pairs of words in the president's Twitter vocabulary (see "Methods").

For each word pair, we modeled diversion as a function of Russia-Mueller media coverage and suppression of subsequent coverage either using two independent OLS models, or using a

| Dependent variable | CJI tweets$_t$ | NYT$^a$ |
|---|---|---|
| | (1) | (2) |
| NYT Brexit$_t$ | 0.028 (0.021) $p = 0.174$ [0.018] $p = 0.126$ | |
| CJI tweets$_{t-1}$ | º | 0.010 (0.058) $p = 0.858$ [0.050] $p = 0.834$ |
| N | 721 | 707 |

**Table 3 Predicting diversionary tweets (CJI) from Brexit media coverage (column 1) and predicting suppression from diversionary tweets (column 2).**

Each model included control variables for each week during the sampling period and long-term time trends as well as the appropriate number of lagged observations for the dependent variable (see "Methods"). Table entries are coefficients (OLS standard errors) [Newey-West standard errors].
$^a$Coverage of Brexit on day $t$.
ºLagged predictor is shown only when of interest.

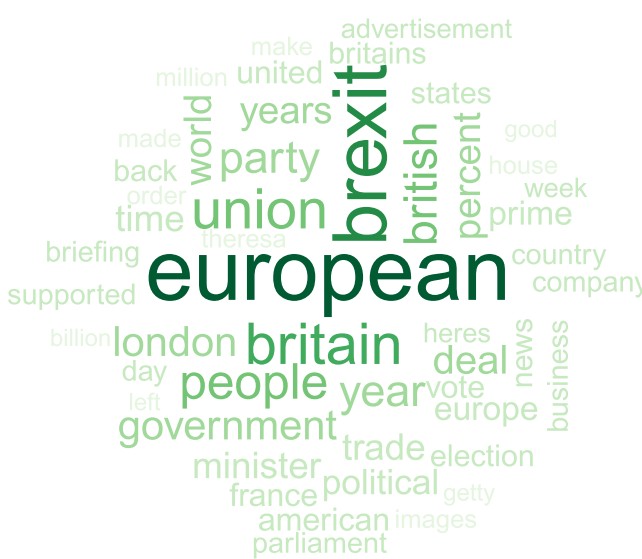

**Fig. 3 Word cloud representation of the top 50 words used in the NYT in their coverage of Brexit.** Size of each word corresponds to its frequency, as does font color. The darker the font, the greater the frequency.

single 3SLS model for both equations simultaneously. The suppression component of the latter model, in turn, either predicted today's coverage from yesterday's tweets or tomorrow's coverage from today's tweets, paralleling the results in panels A and B, respectively, of Table 2.

Figure 4 shows results from the expanded analysis for coverage of Russia-Mueller in the NYT (panels a–c), ABC News (d–f), and the average of both (g–i). Each data point represents a pair of words whose co-occurrence in tweets is predicted by media coverage (position along X-axis) and whose association with subsequent media coverage is also observed (position along Y-axis). Each point thus represents a diversion and a suppression regression simultaneously. The further to the right a point is located, the more frequently the corresponding word pair occurs on days with increasing Russia-Mueller coverage. The lower a point is located, the less Russia-Mueller is covered in the media on the following day as a function of increasing use of the corresponding word pair. If there is no association between the

president's Twitter vocabulary and surrounding media coverage, then all points should lie in the center and largely within the significance bounds represented by the red lines. If there is only diversion, then the point cloud should be shifted to the right. If there is diversion followed by suppression, then a notable share of the point cloud should fall into the bottom-right quadrant.

Figure 4 shows that irrespective of how the data were analyzed (OLS or two variants of 3SLS; columns of panels from left to right), in each instance a notable share of the point cloud sits outside the significance bounds (red lines) in the bottom-right quadrant (summarized further in the Supplementary Table S14). These points represent word pairs that occur significantly more frequently in Donald Trump's tweets when Russia-Mueller coverage increases (i.e., they lie to the right of the vertical red line), and that are in turn followed by significantly reduced media coverage of Russia-Mueller on the following day (i.e., they lie below the horizontal red line). The results are remarkably similar across rows of panels, suggesting considerable synchronicity between the NYT (top row) and ABC (center). The synchronicity is further highlighted in the bottom row of panels, which show the data for the average of the standardized values of coverage in the NYT and ABC.

To provide a chance-alone comparison, the figure also shows the results for the same set of regressions when the Twitter timeline is randomized for each word pair. This synthetic null distribution is represented by the gray contour lines in each panel (the red perimeters represent the 95% boundary). The contrast between the observed data and what would be expected from randomness alone is striking.

To illustrate the linguistic aspects of the observed pattern, the word cloud in Fig. 5 visualizes the words present in all the pairs of words in tweets that occurred significantly more often in response to Russia/Mueller coverage in NYT and ABC News (average of standardized coverage) and that were associated with successful suppression of coverage the next day. The prominence of the keywords from our targeted analysis is immediately apparent. The Supplementary information presents additional quantitative information about those tweets (Table S15).

We performed two control analyses involving placebo keywords to explore whether the observed pattern of diversion and suppression in Fig. 4 reflected a systematic interaction between the president and the media rather than an unknown artifact.

The first control analysis involved NYT Brexit coverage (ABC coverage was too infrequent for analysis, with only a single mention during the sampling period.) For this analysis, articles that contained "Russia" or "Mueller" were excluded to avoid contamination of the placebo coverage by the threatening topic. The pattern for Brexit (Fig. 6) differs from the results for Russia-Mueller. Although Brexit coverage stimulates Twitter activity by President Trump (i.e., points to the right of vertical red line), the word pairs that fall outside the significance boundary tend to be distributed across all four quadrants.

The second control analysis examined other placebo topics using the OLS approach, represented by the NYT keywords "skiing," "football," "economy," "vegetarian," and "gardening" (Fig. 7). The keywords were chosen to span a variety of unrelated domains and were assumed not to be harmful or threatening to the president. For this analysis, the corpus was again restricted to articles that contained neither "Russia" nor "Mueller" to guard against contamination of the coverage by a threatening topic. For most of these keywords, ABC News headlines had zero or one mention only. The exceptions were "football" and "economy," which had 40 and 39 occurrences, respectively. We analyzed ABC News for those two keywords and found the same pattern as for all placebo topics in the NYT. Across the 5 keywords, less than 0.03% of the word pairs (2 out of 7425 for OLS) fell into the

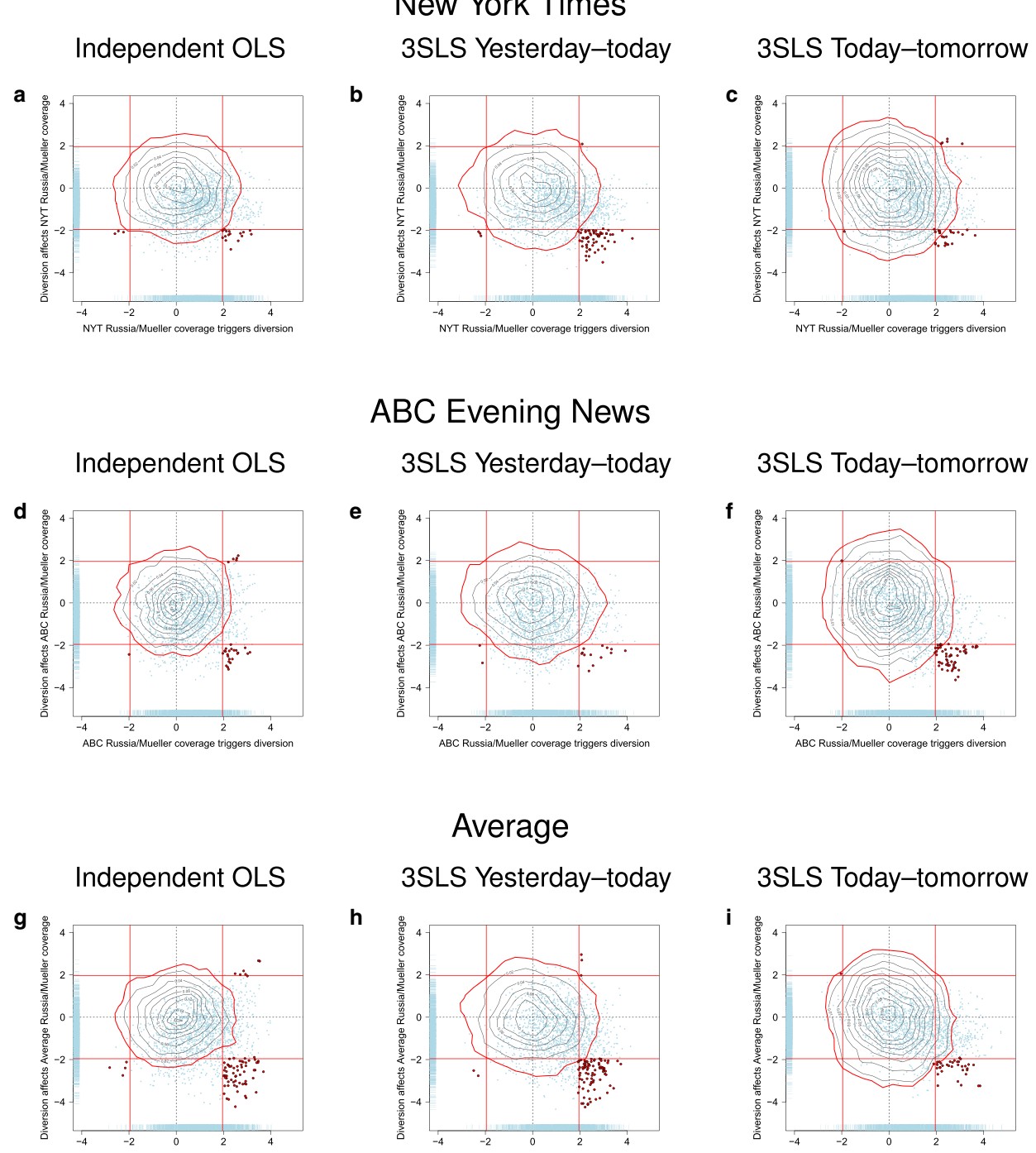

**Fig. 4 Association between Russia-Mueller coverage in the media and diversion, and association between diversion and subsequent Russia-Mueller coverage.** The top row of panels shows the results for the NYT (panels (**a**–**c**)). The center (**d**–**f**) and bottom (**g**–**i**) row of panels show results for ABC News and the average of both media outlets, respectively. The left column of panels (**a**, **d**, **g**) shows results from two independent OLS models, the center column (**b**, **e**, **h**) is for a single 3SLS model in which suppression is modeled by relating yesterday's tweets to today's coverage, and the right column (**c**, **f**, **i**) is a 3SLS model in which suppression is modeled by relating today's tweets to tomorrow's coverage. In each panel, the axes show jittered *t*-values of the regression coefficients for diversion (X-axis) and suppression (Y-axis). Each point represents diversion and suppression for one pair of words in the Twitter vocabulary. Red vertical and horizontal lines denote significance thresholds (±1.96). Word pairs that are triggered by Mueller coverage (*p* < 0.05) and affect subsequent coverage (*p* < 0.05) are plotted in red. The gray contour lines in each panel show the distribution of points obtained if the timeline of tweets is randomized (red perimeter represents 95% cutoff, see "Methods"). The blue rugs represent univariate distributions.

bottom-right quadrant. This number is 40 times smaller than for the parallel NYT analysis for Russia-Mueller.

## Discussion

Our analysis presents empirical evidence that is consistent with the hypothesis that President Trump's use of social media leads to systematic diversion, which in turn may suppress media coverage that is potentially harmful to him. This association was observed after controlling for long-term trends (linear and quadratic), week-to-week fluctuations, and accounting for substantial levels of potential autocorrelations in the dependent variable. The pattern was observed when diversionary topics were chosen a priori to represent the president's preferred political issues, and it also emerged when all possible topics in the president's Twitter vocabulary were considered.

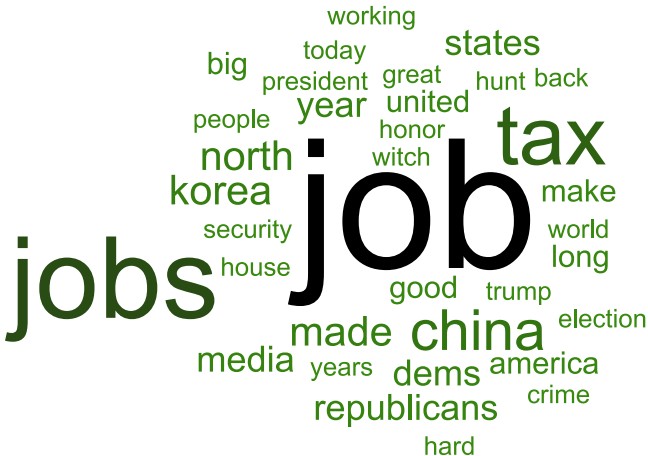

**Fig. 5 Word cloud representation of the word pairs in tweets that were associated with significant diversion and suppression.** The pair structure is ignored in this representation. Size of each word corresponds to its frequency, as does font color. The darker the font, the greater the frequency.

Crucially, in our analysis the diversion and suppression were absent with placebo topics that present no political threat to the president, ranging from Brexit to various neutral topics such as hobbies and food preferences. Our data thus provide empirical support for anecdotal reports suggesting that the president may be employing diversion to escape scrutiny following harmful media coverage and, ideally, to reduce additional harmful media coverage[24,27,43].

Our evidence for diversion is strictly statistical and limited to two media outlets—albeit those commonly acknowledged to be agenda setting in American public discourse—and it is possible that other outlets might show a different pattern. It is also possible that the observed associations do not reflect causal relationships. These possibilities do not detract from the fact, however, that leading media organs in the US are intertwined with the president's public speech in an intriguing manner. We also cannot infer intentionality from these data. It remains unclear whether the president is aware of his strategic deployment of Twitter or acts on the basis of intuition. It is notable in this context that a recent content analysis of Donald Trump's tweets identified substantial linguistic differences between factually correct and incorrect tweets, permitting out-of-sample classifications with 73% accuracy[44]. The existence of linguistic markers for factually incorrect tweets makes it less likely that those tweets represent random errors and suggests that they may be crafted more systematically. In other contexts, deliberate deception has also been shown to affect language use[45].

Questions surrounding intentionality also arise with the recipients of the diversion. It is particularly notable that results consistent with suppression were observed for the *New York Times*, whose coverage has responded strongly to accusations from the president that it spreads fake news, treating those accusations as a badge of honor for professional journalism[46]. The NYT explicitly warns of the impact of Trump's presidency on journalistic standards such as self-censorship, thus curtailing the president's interpretative power. These actions render it unlikely that the NYT would intentionally reduce its coverage of topics that are potentially harmful to the president. The fact that suppression nonetheless occurs implies that important editorial

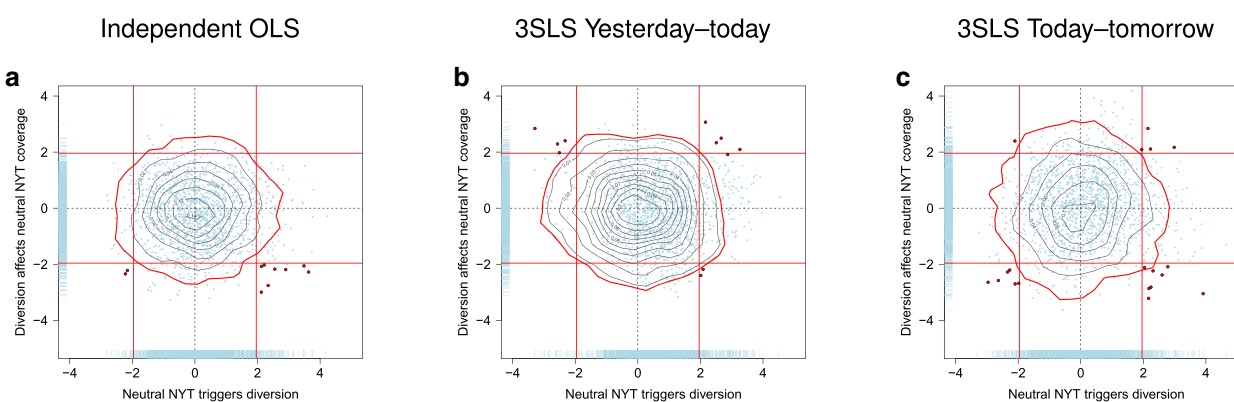

**Fig. 6 The association between Brexit coverage in the NYT and diversion, and the association between diversion and subsequent Brexit coverage.** For this analysis, only NYT articles that did not mention Russia or Mueller were considered ($N = 101, 435$). Panel (**a**) is for two independent OLS models, panel (**b**) is a single 3SLS model in which suppression is modeled by relating yesterday's tweets to today's coverage, and **c** is a 3SLS model in which suppression is modeled by relating today's tweets to tomorrow's coverage. In each panel, the axes show jittered $t$-values of the regression coefficients for diversion (X-axis) and suppression (Y-axis). Each point represents diversion and suppression for one pair of words in the Twitter vocabulary. Red vertical and horizontal lines denote significance thresholds ($\pm 1.96$). Word pairs that are triggered by coverage of the corresponding keyword ($p < 0.05$) and affect subsequent coverage ($p < 0.05$) are plotted in red. The gray contour lines in each panel show the distribution of points obtained if the timeline of tweets is randomized (red perimeter represents 95% cutoff, see "Methods"). The blue rugs represent univariate distributions.

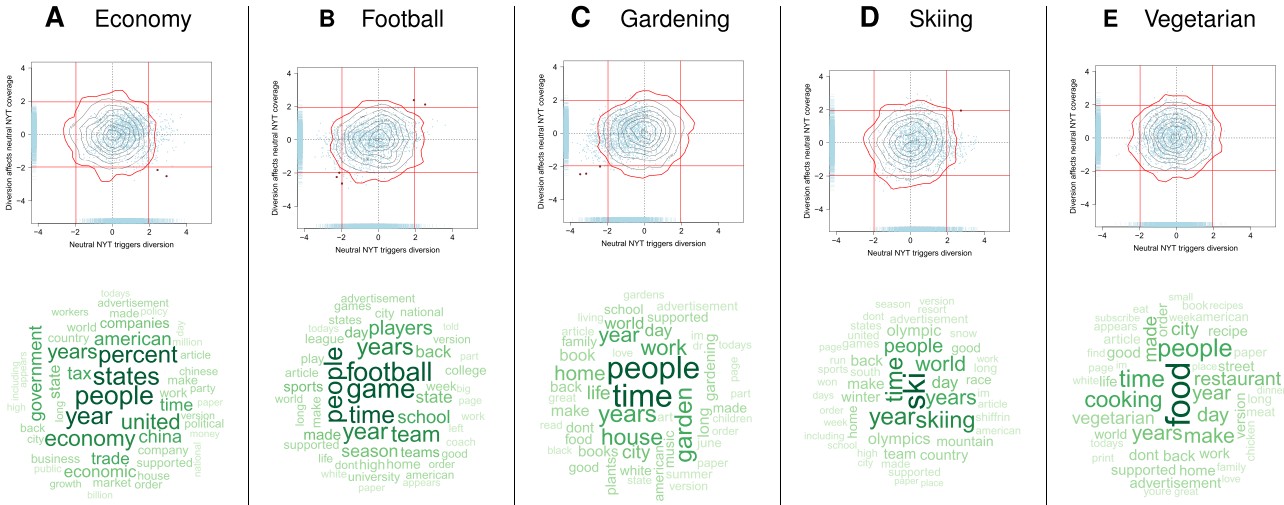

**Fig. 7 Analysis of placebo keywords.** Top row of panels shows the results of the expanded analysis using two independent OLS models for the placebo keywords "economy" (panel (**a**)); "football" (**b**); "gardening" (**c**); "skiing" (**d**); and "vegetarian" (**e**). In each panel, the axes show jittered *t*-values of the regression coefficients for diversion (X-axis) and suppression (Y-axis). Each point represents diversion and suppression for one pair of words in the Twitter vocabulary. Red vertical and horizontal lines denote significance thresholds (±1.96). Word pairs that are triggered by coverage of the corresponding keyword ($p < 0.05$) and affect subsequent coverage ($p < 0.05$) are plotted in red. The gray contour lines in each panel show the distribution of points obtained if the timeline of tweets is randomized (red perimeter represents 95% cutoff, see "Methods"). The blue rugs represent univariate distributions. The word clouds accompanying each plot represent the 50 most frequent words found in the NYT articles selected on the basis of the corresponding keyword.

decisions may be influenced by contextual variables without the editors' intention—or indeed against their stated policies. This finding is not without precedent. Other research has also linked media coverage to extraneous variables that are unlikely to have been explicitly considered by editors or journalists. For example, opinion articles about climate change in major American media have been found to be more likely to reflect the scientific consensus after particularly warm seasons, whereas "skeptical" opinions are more prevalent after cooler temperatures[47].

It is worth drawing links between our results and the literature on the diversionary theory of war[48]. Although it is premature to claim consensual status for the notion that politicians launch wars to divert attention from domestic problems, recent work in history and political science has repeatedly shown an association between domestic indicators, such as poor economic performance or waning electoral prospects, and the use of military force[48]–[51]. Perhaps ironically, this association is particularly strong in democracies[49]. Against this background, the notion that the president's tweets divert attention from inconvenient coverage appears unsurprising.

Finally, we connect our analysis of diversion to other rhetorical techniques linked to President Trump[52]. Each of these presents a rich avenue for further exploration. One related technique involves deflection, which differs from diversion by directly attacking the media (e.g., accusing them of spreading fake news)[25]. Another technique involves pre-emptive framing, by launching information with a new angle[52]. An extension of pre-emptive framing may involve the notion of inoculation, which is the idea that by anticipating inconvenient information, the public may be more resilient to its impact[53]. Inoculation has been shown to be particularly useful in the context of protecting the public against misinformation[54], and it remains to be seen whether pre-emptive tweets by the President may affect the media or the public in their response to an unfolding story.

The availability of social media has provided politicians with a powerful tool. Our data suggest that, whether intentionally or not, President Trump exploits this tool to divert the attention of mainstream media. Further investigation is needed to understand if future US presidents or other global leaders will use the tool in a similar way.

Our findings have implications for journalistic practice. The American media have, for centuries, given much emphasis to the president's statements. This tradition is challenged by presidential diversions in bites of 280 characters. How journalistic practice can be adapted to escape those diversions is one of the defining challenges to the media for the twenty-first century.

## Methods
**Materials**. The sampling period covered 731 days, from Donald Trump's inauguration (20 January 2017) through the end of his 2nd year in office (January 20, 2019). We sampled content items from three sources: (1) all of Donald Trump's tweets from the @realDonaldTrump handle. Tweets that only contained weblinks or were empty after filtering of special characters and punctuation were removed. (2) All of the full-text coverage of the *New York Times* (NYT) available through its online archive. (3) The headlines of all content items covered by the American Broadcasting Corporation's (ABC) *World News Tonight* (daily at 7:30 p.m.). Headlines were obtained from the Vanderbilt Television News Archive (VTNA; https://tvnews.vanderbilt.edu). Summary statistics for the content items are available in the Supplementary information (Table S1).

### Search keys for the targeted analysis
*Threatening media content*. We used the search keys "Russia OR Mueller" to identify media content that was potentially threatening to President Trump. For each day in the sampling period, we counted how many news items in each of our media sources, NYT and ABC, contained one or both of those keywords.

*Diversionary Twitter topics*. We used the search keys "China," "jobs," and "immigration" to identify potentially diverting content in Donald Trump's tweets. For each day in the sampling period, we counted the number of times any of those keywords appeared in the tweets for that day.

**Keywords for the expanded analysis**. The expanded analysis tested all possible pairs of words in Donald Trump's Twitter vocabulary. The vocabulary comprised words that occurred at least 150 times and were not stopwords. (The results are not materially affected if the occurrence threshold is lowered to 100.) Stopwords (such as "the" or "are") are considered unimportant for text analysis. Here we identified stopwords using the SMART option for R package *tm*. We also removed numbers and web links (URLs). Because focus was on diversion, we also excluded "Russia," "Mueller," and "collusion," yielding a final vocabulary of $N = 55$ ($N = 1485$ unique pairs). Each pair was used as a set of keywords in a separate regression analysis. The average number of vocabulary items in a tweet was 2.95 ($s = 2.26$), and we therefore considered two vocabulary words to be sufficient to uniquely identify the topic of a tweet.

**Control variables**. All regression models included at least 106 control variables. A separate intercept was modeled for each of the weeks ($N = 104$) during the sampling period to account for potential endogenous short-term fluctuations. The date (number of days since January 1, 1960), and square of the date, were entered as two additional control variables to account for long-term trends. In addition, lagged variables were included as suggested by a detailed examination of the underlying autocorrelation structures.

**Autocorrelation structure**. We established the autocorrelation structure for each of the variables under consideration by regressing the observations on day $t$ onto the 106 control variables and a varying number of lagged observations of the dependent variable from days $t − 1$, $t − 2$, …, $t − k$. Supplementary Tables S2–S5 report the analyses, which identified the maximum lag ($k$) for each variable. For tweets, the suggested lag was $k = 10$, and for NYT, ABC, and the average it was $k = 28$, $k = 21$, and $k = 28$, respectively. All regressions (OLS and 3SLS; see below) included the appropriate $k$ lagged variables.

**OLS regression models**. The ordinary least-squares (OLS) analyses involved two independently estimated regression models. The first model examined whether Donald Trump's tweets divert the media from threatening media coverage. Given the president's ability to respond nearly instantaneously to media coverage, we considered media coverage and tweets on the same day. That is, we regressed the number of diverting keywords in tweets posted on day $t$ on the number of threatening news items also on day $t$.

The second model examined whether Donald Trump's diverting tweets affected media coverage relating to the potentially threatening topics. This analysis used the number of diverting keywords in the tweets on day $t − 1$ as a predictor for threatening coverage on day $t$. The second model thus assumed a lag between tweets (on day $t − 1$) and an association between those tweets and media coverage (day $t$) to reflect the delay inherent in the news cycle. This model again also included media coverage on day $t − 1$ to capture the inherent momentum of a topic.

All analyses were conducted in R and Stata, either using a robust heteroskedasticity-consistent estimation of the covariance matrix of the coefficient estimates (i.e., function *vcovHC* in the *sandwich* library with the *HC*1 option) or using Newey-West standard errors that are designed to deal with autocorrelated time-series data (*NeweyWest* option for function *coeftest* in R). For a subset of the targeted analysis, the results were reproduced in Stata as a cross-check. All statistical results reported in all tables here and in the Supplementary information are based on two-tailed $t$-tests for the coefficients and are not adjusted for multiple tests.

Because the dependent variables were event counts, a negative binomial regression is often considered to be the most appropriate model for analysis. We report a negative binomial analysis of our main results in the Supplementary information (Table S13). However, negative binomial regressions are known to suffer from frequent convergence problems, and the present analysis was no exception, with the suppression model for ABC failing to converge. In addition, the suppression model for average media coverage cannot be analyzed by a binomial model because the dependent variable (the average of standardized coverage for each media outlet) does not involve counted data. For those reasons, we report the OLS results.

**3SLS regression models**. Given that we expected a tight coupling between diversion and suppression, we applied a three-stage-least-squares (3SLS) regression approach to analyze the coupled system of equations[55]. The 3SLS model jointly predicted Trump's distracting tweets today as a function of adverse coverage on the same day, and Mueller/Russia news today as a function of those tweets yesterday. Because the two components of the 3SLS analysis (diversion and suppression) span different days, we explored another approach in which the suppression was estimated as tomorrow's media coverage as a function of today's tweets. The 3SLS models were fit using Stata's *reg*3 command (called from R via the *RStata* package).

**Noise-only distribution for expanded analysis**. To create a distribution of $t$-values for the coefficients of the expanded analysis when only noise was present, the expanded analysis was repeated for each word pair with the Twitter timeline randomized anew. Randomization disrupts any potential relationship between media coverage and the president's tweets. The density of the bivariate distribution of the $t$-values associated with the regression coefficients was estimated via a two-dimensional Gaussian kernel using the *kde2d* function in the R package *MASS*. The distribution from this randomized analysis should be centered on the origin and be confined largely within the bivariate significance bounds. The contours of the estimated densities are shown in Figs. 4, 6, and 7 as gray lines, with a perimeter line (in red) representing the 95th percentile.

**Reporting summary**. Further information on research design is available in the Nature Research Reporting Summary linked to this article.

## Data availability

A reporting summary is available as a Supplementary Information file. All data is available at https://osf.io/f9bqx/.

## Code availability

All source code for analysis is available at https://osf.io/f9bqx/.

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

## Acknowledgements
The authors are grateful for financial support from the UWA Business School's Future Fund. Preparation of this article was facilitated by grant DP160103596 from the Australian Research Council to U.K.H.E. and S.L.

## Author contributions
S.L. wrote the first draft and conducted the analyses. M.J. obtained and processed the materials and conducted the targeted analysis in Stata. M.J., U.K.H.E., and S.L. all contributed to revision of the paper.

## Competing interests
The authors declare no competing interests.
