## [Peer Review File · Nature Communications]

Editorial Note: Reviewer 4's report contained a comment that was unrelated to the evaluation of the manuscript. This comment has been redacted in the Peer Review File, with Reviewer 4's approval.

Reviewers' comments:

Reviewer #1 (Remarks to the Author):

The manuscript under review seeks to determine whether President Trump successfully diverts media coverage and public attention from an unfavorable topic (Robert Mueller's investigation into his campaign's involvement in Russian interference in the 2016 election) by tweeting about other topics. The topic is interesting and important and not one I have seen studied systematically. However, the empirical analysis does not convincingly establish the claimed findings nor relate it to other cases or theories beyond Trump (e.g., in how presidents like Bill Clinton have sought to make news during scandal). I therefore unfortunately must recommend that the manuscript be declined and that the authors revise the manuscript in preparation for submission to another journal.

The main challenge for the manuscript is that the authors cannot establish a causal relationship between Mueller/Russia news and Trump's tweets on the selected topics or between Trump's tweets and coverage the next day. The former is plausible but the estimated models correlating news and tweets do not establish temporal sequencing (tweets on day t are predicted using coverage on day t conditional on week fixed effects, time, time^2 , and tweets on day $t-1$). The latter predicts coverage on day $t+1$ using tweets on day t conditional on a lagged dependent variable and the same week fixed effects and time variables). To interpret these relationships as causal, one must assume that Russia coverage is randomly assigned conditional on the week/time variables and past values of Trump tweets (for the first type of model) and that Trump tweets are randomly assigned conditional on the week/time variables and past values of Russia coverage (for the second type). (See, e.g., <https://www.mattblackwell.org/files/papers/causal-tscs.pdf>.) Why would these identification assumptions hold? (No evidence is provided for the AR(1) structure of the data either.) To take one simple example, Trump often tweets about jobs on Fridays, when jobs reports come out, and news from the Mueller investigation often came out on Friday afternoons. The authors should thus have day of the week fixed effects. But one could imagine numerous other such stories - for example, if Mueller reporting is a top story, perhaps Trump tweets responding to Mueller reports leads to a massive surge in coverage on day t that then dissipates by $t-1$ in a manner that is correlated with but not caused by Trump's tweets. The authors cannot rule out any such accounts except by assumption and I believe there are many of them. They need an identification strategy or some more precise form of evidence for diversion that goes beyond temporal associations (e.g., showing reporters covering Trump's tweets). The Brexit placebo account is fine but limited; their design does not rule out numerous other potential explanations for the associations they find in coverage of the Mueller report.

Finally, the literature review is quite cursory and the theoretical contribution is underdeveloped. The

authors should at a minimum consider the theoretical motivations for the presumed diversion effect and situate this case among the general set of cases to which they believe these findings apply. This would enable them to better contextualize how their findings might be similar to and/or different from past efforts to divert media attention from unfavorable topics by U.S. presidents or other leaders. They might also consider under what theory of media coverage they would explain the presumed diversion effect. Tweets about jobs have not been especially newsworthy during the Trump administration, especially relevant to the Mueller report. (I'd be more convinced by Trump picking fights with various celebrities and opposition figures - these clashes do generate coverage - but those tweets are hard to identify systematically.)

Reviewer #2 (Remarks to the Author):

This study uses Donald Trump's tweeting behavior as a case study to explore the presidential agenda setting effect. The question, explored by many political communication scholars over the years, is whether and to what extent the president can influence news coverage, and hence, indirectly, public attention, by emphasizing particular topics. The authors in this study go a step further, seeking to link presidential agenda setting power to the notion of diversion. Traditionally, the latter topic has focused on whether the president is able to divert the nation's attention from a bad economy, a scandal, or some other negative information by using military force abroad. But the logic is more or less the same domestically. That is, can a president distract the public from embarrassing or other negative news by emphasizing issues or topics that cast him in a better light?

The authors address these two questions – (1) can the president set the media's agenda, and (2) can the president effectively “change the subject” away from bad news (from his perspective) by talking about something else? – through an empirical investigation of the effects of Trump's Tweets on news coverage of, and public attention to, the Mueller investigation. It is an interesting and, I think, important potential case for exploring these literatures. Moreover, given that, I suspect, future presidents are likely to learn the lesson of Trump and rely increasingly on social media for direct communication with the public, it could prove generalizable, though more time will have to pass before that becomes clear. As of this writing, Trump is unique among American presidents in many ways, one of which is his extensive use of Twitter in lieu of traditional communication paths, like press conferences and briefings.

The basic question is whether Trump was successful in inducing the media to move away from Mueller investigation coverage by talking about other topics, like jobs, China, or immigration, and, relatedly, whether public interest in the Mueller investigation (proxied by Google Trend search patterns) followed a similar path.

The authors conduct two sets of core empirical tests: one using the aforementioned three topics, and the other using all pairs of words appearing in Trump Tweets. In the former analysis, the results appear to indicate that suppression (reducing the media's coverage of the Mueller story, here focusing on ABC World News Tonight and the New York Times) only seems to work with "jobs" in the mix, and we clearly

see the strongest overall effects for "jobs". So, it appears that neither Trump Tweets about China nor about immigration were very successful in suppressing media coverage of Mueller. Nor did they influence public google searches. (I did wonder why not focus on "economy" rather than just "jobs", since Trump seems to talk about various aspects of the economy, including the stock market, manufacturing, etc., not just about "jobs". Perhaps it would be worth casting a somewhat broader net for using the economy as a diversion tactic?)

Of course, what this presentation doesn't tell us is the relative volume of news stories, intensity of media coverage, extent of public interest, or frequency of Tweets, relative to the Mueller story. It would be nice to have a clearer sense of how many stories and how many Tweets are in play, in the form of temporal summary data. Perhaps a line graph of news stories on Mueller juxtaposed against Trump Tweets on topics that seem to matter, along with the Google Trend?

That said, I'm conflicted about the use of google trends data. Because it is "relative volume" and not "actual" volume, it is essentially impossible to know how much people were actually searching. So, you do get some information: was it going up or down from day to day within a given pre-determined search period. But depending on whether or not one has a theoretically justifiable boundary for the search period, the results could be quite misleading. And, as the authors note, for a two-year series, one needs to engage in some methodological manipulations to try to make two periods comparable. The upshot is that I don't think it is worthless. So, its perhaps worth reporting. But it is suggestive at best as an indicator of public interest in a topic, and I think should be characterized as such. (This is not so much a critique of the authors as of Google's odd limitations on use of search trends. But the implication is that I would prefer a bit more circumspect presentation of the Google Trends results.)

The authors wisely also conduct a placebo test using coverage of Brexit. They find, as expected, that Trump's Tweets don't influence subsequent Brexit coverage or public attention the way they do for Mueller coverage. I wonder, though, why exactly that is the case. What is the theoretical mechanism that would lead to Trump's Tweets shifting the media's attention away from Mueller but not away from Brexit? Brexit certainly wasn't in the US news anywhere nearly as regularly as the Mueller investigation. So, there were far fewer stories at play on a regular basis. That alone could explain the absence of influence on the news agenda. I think maybe a better placebo would be something more frequently in the news, like, say, coverage of natural disasters when they arise or other non-political high-profile events? But regardless, I think elaborating on the proposed mechanism at play would be helpful here. Why exactly wouldn't Trump tweeting about jobs result in more coverage of jobs and relatively less coverage of Brexit, unless there would otherwise have not been sufficient coverage of Brexit to generate a statistically significant shift. Same question regarding the Google Trend.

Turning to the core theories underlying this study, there is a small but not trivial literature in political communication on "who leads whom": the press, the public, or politicians/presidents? For instance, Canes-Wrone wrote an influential book on the question regarding presidents vs. public opinion (see also Shapiro & Jacobs 2000 and others). There are a fair number of studies on the president vs. the media as

well (e.g., Edwards & Wood 1999, Wood & Peak 1998, Baum and Potter 2008, Bartels 1996, Wanna and Ford 1994, and others) I don't see any of this literature cited in this paper, which seems odd to me. There is but one site to the agenda setting literature as well, and none at all to the diversionary war literature, which is a close cousin of this study. I don't mention all these particular papers because I think the authors need to cite them all. But I do think this paper, with relatively minor tweaks, would have stronger theoretical bite and more generalizable implications if it more explicitly recognized that this study is an extension of what others have been attempting to do for decades, with admittedly imperfect success. This is a very cool empirical finding, and so I think ultimately publishable. But it is fundamentally an extension of prior empirical findings -- using better and more data, and more current communication technology and strategies -- that presidents are able to influence the media's short-term agenda by talking about a different topic.

That said, to reiterate, the carrying of the traditional literature over to a social media context (twitter) is by itself interesting and, I think, a great empirical case study of this hypothesis.

One thing the authors didn't really do is explore the ubiquity of Trump's ability to divert the press and/or the public. They do this a little bit by comparing china vs. immigration vs. jobs. But what that left me wondering is whether Trump is actually able to just Tweet away about anything, or whether it needs to be something about which the public is highly interested and/or there is significant news that the media could be covering. And can such Tweets divert from "any" topic? For instance, would it work with impeachment as well as with Mueller? Why or why not? As I noted, it seems that in the first analysis, only jobs really succeeded in diverting press and public attention very much. So, I wonder if it would be possible to add to the second analysis some sort of hierarchy of word-pair efficacy in diversion, at the end of which we might be able to say something about whether "anything works" or "almost anything" or "only a few things" or "almost nothing". I would love to see some sort of graphical representation of the top set of word-pairs in descending order, or something like that.

Similarly, it would be interesting to see if the interaction between the choice of Tweet topics and exogenous circumstances or events is significant. Does Trump talking about jobs on the day the monthly jobs report is released work better as a Mueller distraction than talking about jobs on other days? Does it depend on the existence of some salient news on the dimension upon which Trump is Tweeting? Or can he just say "jobs" at any time with similar effect?

In sum, I like this study and think it's a good candidate for publication. But I do have some questions about it and suggestions for revisions that I think might strengthen it. So, I would like to see the authors address as many of these as they can.

Reviewer #3 (Remarks to the Author):

NCOMMS-19-31135 - Donald Trump and strategic diversion in the age of Twitter

The argument of this paper is compelling and well worth studying. We know that agenda setting is possible even when persuasion is not, and agenda setting often has downstream consequences, such as affecting the emphasis on different issues in the media and by public policy makers. That said, I think the analysis in the paper can be greatly improved. I would like to see a revised version published, but I'm afraid I'm about to make a lot of suggestions for substantial additional work.

When Google Trends got started, everyone -- including Google -- was worried about it being used for academic research. As I recall, when it originally appears, Google's website explicitly said that it was not suitable to write a dissertation based on it. Since then we've all become accustomed to it, but that does not mean it is any more valid. I think the authors should address this issue directly. For any measure, readers must know the chain of evidence from the world we're studying to the data we have. What do we know in this case, since Google's algorithms are hidden from us, and it isn't clear what their reporting rules are. I don't see anything in the Google Trends help files presently that indicates much of anything useful in this regard. Do the authors have some way to justify their measure (other than that others are using it)?

If the authors can justify the issues in the previous paragraph, then next up is to justify their use of the very few keywords they chose. Do these keywords measure the intent of users in the way that they expect? What way do they expect? Have they looked at what websites come up (in different parts of the country) when different types of users search for these keywords? That is, we need to first identify the quantity of interest in order to evaluate the measure. In this case, I would think that the quantity of interest must have to do with either the types of websites that users want to find or the ones they do find by performing a Google search. If so, then the authors could do a study to ascertain whether their measure has good statistical properties. Is the measure (i.e., Google Trends for each keyword or for all together if they like) an unbiased estimator of the quantity of interest? And we need to figure out both precision and recall -- or false positives and false negatives. How often does searching for "China" or "jobs" indicate that the user is looking for the same topic Trump sparked in his Tweets? And also, how many are looking for the topic of interest but are missed by the same keywords? To do the latter, the authors would need to identify a large group of keywords representing their theoretical concept.

Another issue is that the authors need to explain what data were collected, the structure of the data, and basic facts like the unit of analysis, what justifies their timeframe, and how many observations they have -- all before reporting their regression results. Otherwise, there's no motivation to read the results at all. Yes, I realize that methods comes after, but what is the point of me saying that my evaluation of this paper is 197.43 unless you know the scale first?

A fundamental issue is what causal identification assumptions the authors are claiming. I do not see a discussion of this critical issue. I would think that the results are all afflicted by post-treatment bias, at a minimum, almost by definition (if I'm guessing what the authors are doing correctly). And there would seem to be other issues, but the burden of proof is on the author here and I don't see a case being made.

The NY Times is a highly liberal news outlet. I would think the authors should match it with a reasonable conservative outlet, like the WSJ. It would make their results more compelling if they found a similar result.

Do the statistical models fit the data? The outcome is a count (the number of diverting keywords), which would normally be analyzed by event count models rather than linear regression. These data (and models) tend to be heteroskedastic, which would be ignored by linear regression. And probably some of the predicted values of the regression will be negative, which of course makes no sense.

Response to reviewers

(Original text in courier font, replies in *italics*.)

Reviewers' comments:

Reviewer #1 (Remarks to the Author):

The manuscript under review seeks to determine whether President Trump successfully diverts media coverage and public attention from an unfavorable topic (Robert Mueller's investigation into his campaign's involvement in Russian interference in the 2016 election) by tweeting about other topics. The topic is interesting and important and not one I have seen studied systematically. However, the empirical analysis does not convincingly establish the claimed findings nor relate it to other cases or theories beyond Trump (e.g., in how presidents like Bill Clinton have sought to make news during scandal). I therefore unfortunately must recommend that the manuscript be declined and that the authors revise the manuscript in preparation for submission to another journal.

The main challenge for the manuscript is that the authors cannot

establish a causal relationship between Mueller/Russia news and Trump's tweets on the selected topics or between Trump's tweets and coverage the next day. The former is plausible but the estimated models correlating news and tweets do not establish temporal sequencing (tweets on day t are predicted using coverage on day t conditional on week fixed effects, time , time^2 , and tweets on day $t-1$). The latter predicts coverage on day $t+1$ using tweets on day t conditional on a lagged dependent variable and the same week fixed effects and time variables). To interpret these relationships as causal, one must assume that Russia coverage is randomly assigned conditional on the week/time variables and past values of Trump tweets (for the first type of model) and that Trump tweets are randomly assigned conditional on the week/time variables and past values of Russia coverage (for the second type). (See, e.g., <https://www.mattblackwell.org/files/papers/causal-tscs.pdf>.) Why would these identification assumptions hold? (No evidence is provided for the AR(1) structure of the data either.) To take one simple example, Trump often tweets about jobs on Fridays, when jobs reports come out, and news from the Mueller investigation often came out on Friday afternoons. The authors should thus have day of the week fixed effects. But one could imagine numerous other such stories - for example, if Mueller reporting is a top story, perhaps Trump tweets responding to Mueller reports leads to a massive surge in coverage on day t that then dissipates by $t-1$ in a manner that is correlated with but not caused by Trump's tweets. The authors cannot rule out any such accounts except by assumption and I believe there are many of them. They need an identification strategy or some more precise form of evidence for diversion that goes beyond temporal associations (e.g., showing reporters covering Trump's tweets).

We appreciate and generally agree with the reviewer's concern regarding the identification of causality, and we have dedicated much effort to addressing the reviewer's thoughtful comments. In the following, we address the two principal points separately.

Autocorrelational structure

First, we address the tacit assumption of an AR(1) structure that characterized our first submission. We agree that a more thorough treatment of the underlying lag structure, as well as more empirical evidence, was required. In the new Tables S1-S3 in the supplementary material, we show the results from regressing the respective dependent variable on their lagged variables, including our control variables of week-fixed effects and time trends (linear and squared).

To briefly summarize those results, for President Trump's tweets of the critical diversionary keywords "China", "Jobs", and "Immigration" (CJI from here on), lags beyond 10 days become largely statistically irrelevant by conventional significance criteria (Table S1, column 5). If we include 10 lagged values of the dependent variable, we replicate our main model that predicts Trump's CJI tweets: These results are displayed in the new Table 1 in the main text, in columns (1) to (3). The table shows that NYT coverage, ABC news segments, and the combination of both (formed by averaging the standardized values of both variables) are associated with an increased number of CJI tweets on the same day.

This result emerges with conventional OLS and also when we employ Newey-West standard errors that are designed to deal with autocorrelated time-series data (command `newey` in Stata and "NeweyWest" option for `coefest` in R). The results are consistent across both types of standard error, attesting to the robustness of the relationship: irrespective of whether we consider NYT, ABC, or both combined, coverage of Russia-Mueller is associated with more CJI tweets by Donald Trump.

Turning to the lag structure of NYT Mueller/Russia news, Table S2 shows that the data exhibit strong autocorrelations for more than 20 days. Thus, in column 4 of Table 1 in the main text we re-estimate the suppression of NYT coverage of Mueller/Russia through CJI tweets on the previous day, controlling for 28 lagged variables of the dependent measure. Our principal result again prevails, albeit at a reduced level of statistical significance ($p < .10$). The same result emerges for ABC News segments, with Table S3 exploring the lag structure and column 5 of Table 1 re-estimating ABC suppression with 21 lagged variables. Finally, when we combine both media outlets by averaging their standardized values, the model including autocorrelations is significant at conventional levels ($p < .05$; column 6 in Table 1 for both types of standard error).

In summary, a detailed exploration of the lag structure of our data does not alter the principal results of our targeted analysis. Likewise, our revised expanded analysis—described in detail in the next section—also returns the same main result as before but after addition of the lagged variables and using a far more rigorous model that we outline next.

Endogeneity concerns

Second, we address the reviewer's main point regarding endogeneity. The reviewer is entirely correct, of course, in stating that we do not have a bulletproof identification strategy. In principle, endogeneity may have accounted for the observed correlation between CJI tweets and Mueller/Russia news (our first result) and/or the negative link between Trump's CJI tweets and a subsequent reduction in this coverage (our second finding). Thus, we acknowledge that endogeneity is a valid concern and we agree that our initial manuscript did not discuss these issues carefully enough. We now offer several reasons to address these concerns. To do so, we turn to the three main types of endogeneity in turn: (1) Measurement error; (2) reverse causality; and (3) omitted variables.

(1) Measurement error. We find it implausible that measurement error plays a significant role in our setting. We draw from the universe of all NYT articles, ABC News segments, and Trump tweets in our sampling period. The only room for measurement error would be that we might miss some articles, news segments, or tweets (perhaps because our keywords did not fully catch all relevant items). It is difficult for us to imagine a scenario in which this could introduce a systematic bias in only one direction that would fundamentally inflate our estimates.

(2) Reverse causality. For the first model (diversion), it is logically difficult to see how Trump's tweets on China, jobs, or immigration could lead to more news about the Mueller investigation on the same day. The first reason for this is that given the lag time of news reports, there is little opportunity for a tweet to generate substantial coverage within the same 24-hour period. Moreover, even putting aside that timing constraint, we find it difficult to imagine how the media would systematically report more on Mueller/Russia because President Trump tweeted on China, jobs, or immigration—by what logic would the media respond in this manner? The reverse, of course, is not only plausible but motivated our analysis, namely the hypothesis that Trump would divert with irrelevant topics when confronted by uncomfortable media coverage.

For the second model (suppression), the fact that we predict news today with Trump's tweets from yesterday (because of the reasons just cited, namely that reporters cannot respond instantaneously) reduces concerns about reverse causality.¹ One might entertain the possibility that Trump tweets about CJI pre-emptively, in expectation of Mueller coverage tomorrow. However, that would, if anything, work against us detecting the suppression effect. If Trump tweeted in expectation, we should see more Trump tweets on days prior to increased Mueller coverage, thus creating a positive association. However, our result is a statistically significant negative relationship which is the opposite of the hypothetical anticipation scenario. Thus, if anything, such dynamics would suggest our findings to constitute a lower bound, stacking the deck against our observed negative association.

¹ We also predict suppression tomorrow from tweets today in a 3SLS model (explained next) but that also does not permit reverse causality because a later event is predicted by an earlier event.

(3) *Omitted variables*. This presents the most serious threat to causal identification in our setting. As the reviewer notes, it is virtually impossible to systematically incorporate all variables that could simultaneously affect Trump's CJI tweets and news coverage of the Mueller/Russia investigation, thereby artifactually giving rise to the observed association. We triangulate this problem in several ways, and we believe that the sum total of our efforts addresses the omitted-variables explanation as much as we possibly can.

For our first model (diversion), we test for the lurking effects of omitted variables with the Oster test (Oster, E., 2019. *Unobservable selection and coefficient stability: Theory and evidence. Journal of Business & Economic Statistics*, 37, 187-204). The Oster test explores the following hypothetical question: How important must the set of unobservable omitted variables be in order to “explain away” the regression coefficient that is estimated in the presence of those unknown omitted variables? The achieved (i.e., observed) R^2 s in our analysis are .36 for the diversion models based on NYT, ABC, and the combined media (new Table 2). Thus, around 36% of the variation in the dependent variable (Trump's CJI tweets) are explainable by our regressors, including the media coverage of Mueller/Russia.

The Oster test allows the researcher to choose a hypothetical R^2 value that they believe to be the ultimately explainable variation in the dependent variable, assuming all unobservable variables were available. The test then calculates how large a “ δ ” (the extent to which unobservables explain variance relative to that explained by the observed variables) would need to be in order to render the observed variables unimportant given the assumed maximum-attainable R^2 . The test provides formal guidance for interpreting the values of δ .

The new Table 2 displays the results for different assumed maximum-attainable values of R^2 (R_{max} from here on, following Oster's notation). Those hypothetical R_{max} values are 0.75, 0.8, and 1. This represents the assumptions that, if a researcher had all the data in the world, we would be able to explain 75%, 80%, or 100% of Trump's CJI tweets. The higher the assumed maximum-attainable value, the more likely it is that unobservables contribute to the effect (because more remains to be explained relative to the actually observed R^2). It appears highly implausible to us that all of the variation in Trump's tweets might be systematically explainable: the hypothesized R_{max} of 1 thus represents an extreme test of the role of unobserved variables.

We computed the values of δ for the three assumed values of R_{max} . Oster (2019) suggests that any value of δ above 1 attests to the robustness of a model because it “suggests the observables are at least as important as the unobservables” (p. 196). Table 2 shows that $\delta > 1$ for all three models (NYT, ABC, both combined) with a single exception: for NYT and $R_{max} = 1$, δ falls below 1. However, as noted, $R_{max} = 1$ constitutes an extreme and unlikely assumption because it implies that none of Trump's tweets arise from a statistically random process.

In sum, we believe that the results of the Oster test make it unlikely that omitted variables are able to explain our first result in its entirety. We are therefore reasonably confident—without being able to prove causality—that we have established the use of diversion in Trump's tweeting behaviour. Nevertheless, the narrative of our new version carefully tries to reflect endogeneity concerns and not overstate our findings.

Turning to omitted variables in our second result (suppression), we address several objections in turn. The first objection by Reviewer 1 was perhaps Trump tweets responding to Mueller reports leads to a massive surge in coverage on day t that then dissipates by $t+1$ in a manner that is correlated with but not caused by Trump's tweets. This is an important point. However, our new models control for Mueller/Russia news on day t , as well as on several days prior to that (see the preceding treatment of autocorrelations). Thus, what we identify in the revised models is a negative relationship between Trump's CJI tweets on day t and Mueller/Russia news coverage on day $t+1$ that accounts for past Mueller/Russia coverage. Our revised models therefore put to rest this particular objection.

Moreover, for (an) omitted variable(s) to explain our observed suppression, we would need to postulate something unobservable that leads to more (fewer) Trump tweets on CJI on day t and less (more) Mueller/Russia coverage on day $t+1$. As Reviewer 1 correctly suggests, one may be able to propose narratives that fulfil those assumptions. What makes our case more complicated is the fact that our two results (diversion and suppression) are tightly coupled by our hypothesis: we hypothesize that the diversions work, that is, the CJI-related tweets that are aimed to divert may actually accomplish precisely that and hence suppress media coverage. This conceptual complication, however, also offers an avenue towards a solution: In essence, the paired nature of our expectations and findings, namely that diversion is followed by suppression, would suggest a joint estimation of both equations. We therefore estimated three-stage-least-squares regressions (Zellner, A., & Theil, H., 1962, *Three-stage least squares: simultaneous estimation of simultaneous equations*, *Econometrica*, 30, 54-78; 3SLS from here on) to jointly predict Trump's CJI tweets today as a function of adverse coverage on the same day, and Mueller/Russia news on the subsequent day as a function of those tweets.

The new Table 3 displays the 3SLS results for the targeted analysis, using two ways to model the temporal offset between tweets and subsequent, potentially suppressed, media coverage. In panel A, we used yesterday's tweets to predict today's coverage. This parallels the OLS analyses. In panel B, we seek to predict tomorrow's coverage from today's tweets. The pattern is remarkably similar across both panels: There are strong and nearly-uniformly significant coefficients of adverse media coverage predicting diversion. Conversely, all coefficients for suppression are negative, as in the OLS analysis (Table 1), although their significance levels are attenuated compared to the OLS analysis.

In our view, the 3SLS results further diminish the likelihood of an artifactual explanation: for omitted variables to explain the observed joint pattern of diversion and suppression, those variables would have to simultaneously explain a positive association between two variables on the same day and a negative association from one day to the next across two different intervals—namely from yesterday to today as well as from today to tomorrow. And those omitted variables would have to exert their effect in the presence of more than 100 other control variables in the models. We consider this possibility to be unlikely.

To buttress our conclusion further, the expanded analysis using both approaches to 3SLS (new Figure 3) shows the same pattern of effects as reported in the original manuscript. We therefore suggest that we have done everything possible to address endogeneity concerns in our setting. Of course, we are conscious of the fact that, in the absence of a controlled experiment that manipulates the President's tweeting behaviour, we cannot fully identify causal effects, and any interpretation of our findings will have to keep that in mind. Nevertheless, we think these findings are important in light of current political developments and are consonant with prior theorizing that until now had to remain largely speculative.

The Brexit placebo account is fine but limited; their design does not rule out numerous other potential explanations for the associations they find in coverage of the Mueller report.

Brexit is only one of several placebo topics that we have examined. Across a wide range of keywords covering sports (e.g., "basketball"), food (e.g., "vegetarian"), and other political topics (e.g., "economy"), we never observe the distinct pattern of diversion and suppression suggested by our results pertaining to the Russia/Mueller coverage (see new Figure 4). This does not conclusively rule out alternative explanations for the Russia/Mueller results, but it makes those results even more distinctive. The challenge for an alternative explanation would be to explain why a spurious or noncausal association arises only for Russia/Mueller but not across a variety of other topics.

Finally, the literature review is quite cursory and the theoretical contribution is underdeveloped. The authors should at a minimum

consider the theoretical motivations for the presumed diversion effect and situate this case among the general set of cases to which they believe these findings apply. This would enable them to better contextualize how their findings might be similar to and/or different from past efforts to divert media attention from unfavorable topics by U.S. presidents or other leaders. They might also consider under what theory of media coverage they would explain the presumed diversion effect. Tweets about jobs have not been especially newsworthy during the Trump administration, especially relevant to the Mueller report.

Our initial treatment of the literature was cursory because we wanted to keep the paper relatively brief and to the point. We have now provided considerable additional discussion of the literature. Of course, the revision has grown in length as a result.

(I'd be more convinced by Trump picking fights with various celebrities and opposition figures - these clashes do generate coverage - but those tweets are hard to identify systematically.)

We agree that "picking fights" may be another diversionary strategy. Indeed, we have reported an anecdotal instance of this in the literature (the "Hamilton" Broadway play controversy; Lewandowsky, S., Ecker, U. K. H., & Cook, J. (2017). Beyond Misinformation: Understanding and coping with the post-truth era. Journal of Applied Research in Memory and Cognition, 6, 353-369. DOI: 10.1016/j.jarmac.2017.07.008). However, as the reviewer suspects, those instances are very difficult to identify systematically and we therefore did not include those tweets in the present project.

Reviewer #2 (Remarks to the Author):

This study uses Donald Trump's tweeting behavior as a case study to explore the presidential agenda setting effect. The question, explored by many political communication scholars over the years, is whether and to what extent the president can influence news coverage, and hence, indirectly, public attention, by emphasizing particular topics. The authors in this study go a step further, seeking to link presidential agenda setting power to the notion of diversion. Traditionally, the latter topic has focused on whether the president is able to divert the nation's attention from a bad economy, a scandal, or some other negative information by using military force abroad. But the logic is more or less the same domestically. That is, can a president distract the public from embarrassing or other negative news by emphasizing issues or topics that cast him in a better light?

The authors address these two questions - (1) can the president set the media's agenda, and (2) can the president effectively "change the subject" away from bad news (from his perspective) by talking about something else? - through an empirical investigation of the effects of Trump's Tweets on news coverage of, and public attention to, the Mueller investigation. It is an interesting and, I think, important potential case for exploring these literatures. Moreover, given that, I suspect, future presidents are likely to learn the lesson of Trump and rely increasingly on social media for direct

communication with the public, it could prove generalizable, though more time will have to pass before that becomes clear. As of this writing, Trump is unique among American presidents in many ways, one of which is his extensive use of Twitter in lieu of traditional communication paths, like press conferences and briefings.

The basic question is whether Trump was successful in inducing the media to move away from Mueller investigation coverage by talking about other topics, like jobs, China, or immigration, and, relatedly, whether public interest in the Mueller investigation (proxied by Google Trend search patterns) followed a similar path.

The authors conduct two sets of core empirical tests: one using the aforementioned three topics, and the other using all pairs of words appearing in Trump Tweets. In the former analysis, the results appear to indicate that suppression (reducing the media's coverage of the Mueller story, here focusing on ABC World News Tonight and the New York Times) only seems to work with "jobs" in the mix, and we clearly see the strongest overall effects for "jobs". So, it appears that neither Trump Tweets about China nor about immigration were very successful in suppressing media coverage of Mueller. Nor did they influence public google searches. (I did wonder why not focus on "economy" rather than just "jobs", since Trump seems to talk about various aspects of the economy, including the stock market, manufacturing, etc., not just about "jobs". Perhaps it would be worth casting a somewhat broader net for using the economy as a diversion tactic?)

The original Table 4 (now Table 6) has addressed this question because "economy" is not among the diversionary vocabulary. We have further expanded on the "economy" theme by reporting another placebo analysis using that term and finding that it did not trigger any systematic diversion or suppression. We interpret this to mean that the economy is simply too broad a term, in both media and the President's tweets, to be used as, or to trigger a, diversion.

Of course, what this presentation doesn't tell us is the relative volume of news stories, intensity of media coverage, extent of public interest, or frequency of Tweets, relative to the Mueller story. It would be nice to have a clearer sense of how many stories and how many Tweets are in play, in the form of temporal summary data. Perhaps a line graph of news stories on Mueller juxtaposed against Trump Tweets on topics that seem to matter, along with the Google Trend?

This is an excellent idea but it is difficult to present graphically because the absolute numbers of events on each day during the two-year period is small and the relationship between the two variables would be difficult to represent graphically as two time series. Instead, we have plotted the estimates of the two effects of interest (CJI diversion and subsequent suppression) together with the residuals to give an impression of the strength of the effect. This is shown in the new Figure 2.

That said, I'm conflicted about the use of google trends data. Because it is "relative volume" and not "actual" volume, it is essentially impossible to know how much people were actually searching. So, you do get some information: was it going up or down from day to day within a given pre-determined search period. But depending on whether or not one has a theoretically justifiable

boundary for the search period, the results could be quite misleading. And, as the authors note, for a two-year series, one needs to engage in some methodological manipulations to try to make two periods comparable. The upshot is that I don't think it is worthless. So, its perhaps worth reporting. But it is suggestive at best as an indicator of public interest in a topic, and I think should be characterized as such. (This is not so much a critique of the authors as of Google's odd limitations on use of search trends. But the implication is that I would prefer a bit more circumspect presentation of the Google Trends results.)

We have given the Google Trends analysis much consideration, and even though we believe that we can respond to this critique, as well as Reviewer 3's more detailed criticisms, we have decided to remove that Google Trends analysis altogether. This tightened the focus of the paper without losing too much information and allowed us to streamline our discussion.

The authors wisely also conduct a placebo test using coverage of Brexit. They find, as expected, that Trump's Tweets don't influence subsequent Brexit coverage or public attention the way they do for Mueller coverage. I wonder, though, why exactly that is the case. What is the theoretical mechanism that would lead to Trump's Tweets shifting the media's attention away from Mueller but not away from Brexit? Brexit certainly wasn't in the US news anywhere nearly as regularly as the Mueller investigation. So, there were far fewer stories at play on a regular basis. That alone could explain the absence of influence on the news agenda. I think maybe a better placebo would be something more frequently in the news, like, say, coverage of natural disasters when they arise or other non-political high-profile events?

As noted in our response to Reviewer 1, we have extended the placebo analyses to other topics (including "economy") that have received far more coverage in the U.S. than Brexit. Specifically, we used the NYT keywords "skiing", "football", "economy", "vegetarian", and "gardening" (ABC News headlines were only sufficient for "football" and "economy"). These keywords were chosen to span a variety of unrelated domains and were assumed not to be harmful or threatening to the president. For the placebo analyses, the corpus was restricted to articles that contained neither "Russia" nor "Mueller" to guard against contamination of the coverage by a threatening topic. Across the 5 keywords, only 0.03% of the word pairs (2 out of 7,425) fell into the bottom-right quadrant. This number is roughly 40 times smaller than for Russia-Mueller.

But regardless, I think elaborating on the proposed mechanism at play would be helpful here. Why exactly wouldn't Trump tweeting about jobs result in more coverage of jobs and relatively less coverage of Brexit, unless there would otherwise have not been sufficient coverage of Brexit to generate a statistically significant shift.

Even if one postulates that there is insufficient coverage of Brexit to detect shifts statistically, this criticism does not apply to the other placebo keywords, in particular "economy." Nonetheless, we failed to observe any systematic effects in the analysis of "economy".

Same question regarding the Google Trend.

We have removed the Google Trends analysis, so this point no longer applies. Nonetheless, for the sake of intellectual curiosity, our response to this point would be that while one would expect the president to be aware of content in the major national media, it is less likely that he would be aware of public opinion or interest – for which Google Trends was a proxy – on a daily basis. Indeed, if the president's tweeting behaviour were a function of Google Trends, this would concern us because a priori we would not expect that effect. The converse, however, is not true: the president's tweets may well divert the public's attention, perhaps mediated by an effect on media coverage, which is precisely what we found.

Turning to the core theories underlying this study, there is a small but not trivial literature in political communication on "who leads whom": the press, the public, or politicians/presidents? For instance, Canes-Wrone wrote an influential book on the question regarding presidents vs. public opinion (see also Shapiro & Jacobs 2000 and others). There are a fair number of studies on the president vs. the media as well (e.g., Edwards & Wood 1999, Wood & Peak 1998, Baum and Potter 2008, Bartels 1996, Wanna and Ford 1994, and others) I don't see any of this literature cited in this paper, which seems odd to me.

We agree that this came across as odd, but we were keen to keep the paper short and concise, following the style of the majority of articles in this journal. As brevity does not appear to be a concern, we have considerably expanded the discussion of agenda-setting more generally. We now cite much additional literature, including some of the papers provided by the reviewer. Thank you for your suggestions.

There is but one site to the agenda setting literature as well, and none at all to the diversionary war literature, which is a close cousin of this study.

We had not thought of this angle and are grateful to the reviewer for pointing this out to us. This is indeed very interesting. We have explored this literature and are now referring to relevant work.

I don't mention all these particular papers because I think the authors need to cite them all. But I do think this paper, with relatively minor tweaks, would have stronger theoretical bite and more generalizable implications if it more explicitly recognized that this study is an extension of what others have been attempting to do for decades, with admittedly imperfect success. This is a very cool empirical finding, and so I think ultimately publishable. But it is fundamentally an extension of prior empirical findings -using better and more data, and more current communication technology and strategies -- that presidents are able to influence the media's short-term agenda by talking about a different topic.

We agree and hope that the revision has allayed those concerns.

That said, to reiterate, the carrying of the traditional literature over to a social media context (twitter) is by itself interesting and, I think, a great empirical case study of this hypothesis.

One thing the authors didn't really do is explore the ubiquity of Trump's ability to divert the press and/or the public. They do this

a little bit by comparing china vs. immigration vs. jobs. But what that left me wondering is whether Trump is actually able to just Tweet away about anything, or whether it needs to be something about which the public is highly interested and/or there is significant news that the media could be covering. And can such Tweets divert from "any" topic? For instance, would it work with impeachment as well as with Mueller? Why or why not? As I noted, it seems that in the first analysis, only jobs really succeeded in diverting press and public attention very much.

Our expanded analysis looked at all possible pairs of content words in tweets and found that a subset of those managed to divert and suppress.

So, I wonder if it would be possible to add to the second analysis some sort of hierarchy of word-pair efficacy in diversion, at the end of which we might be able to say something about whether "anything works" or "almost anything" or "only a few things" or "almost nothing". I would love to see some sort of graphical representation of the top set of word-pairs in descending order, or something like that.

This is an interesting idea although there are several candidate metrics for this. We have chosen to use the average of the diversion and suppression coefficients (actually the t-values, to be consistent with the Y-axis of the corresponding figure) to rank-order the word pairs in what is now Table 6 (for the average of ABC and NYT).

Similarly, it would be interesting to see if the interaction between the choice of Tweet topics and exogenous circumstances or events is significant. Does Trump talking about jobs on the day the monthly jobs report is released work better as a Mueller distraction than talking about jobs on other days? Does it depend on the existence of some salient news on the dimension upon which Trump is Tweeting? Or can he just say "jobs" at any time with similar effect?

We have delved into this arena during a search for clearly exogenous events that might be driving tweets (and media coverage) without concerns about endogeneity. Unfortunately, the candidate events were found to be insufficiently frequent to permit a systematic empirical analysis.

In sum, I like this study and think it's a good candidate for publication. But I do have some questions about it and suggestions for revisions that I think might strengthen it. So, I would like to see the authors address as many of these as they can.

We appreciate the reviewer's positive stance and are particularly grateful for the pointers to additional literature. We hope that we have addressed all the comments.

Reviewer #3 (Remarks to the Author):

The argument of this paper is compelling and well worth studying. We know that agenda setting is possible even when persuasion is not, and agenda setting often has downstream consequences, such as affecting the emphasis on different issues in the media and by public policy makers. That said, I think the analysis in the paper

can be greatly improved. I would like to see a revised version published, but I'm afraid I'm about to make a lot of suggestions for substantial additional work.

When Google Trends got started, everyone -- including Google -- was worried about it being used for academic research. As I recall, when it originally appears, Google's website explicitly said that it was not suitable to write a dissertation based on it. Since then we've all become accustomed to it, but that does not mean it is any more valid. I think the authors should address this issue directly. For any measure, readers must know the chain of evidence from the world we're studying to the data we have. What do we know in this case, since Google's algorithms are hidden from us, and it isn't clear what their reporting rules are. I don't see anything in the Google Trends help files presently that indicates much of anything useful in this regard.

We appreciate the reviewer's concern. After thinking about this for considerable time we decided to respond to this concern by removing the Google Trends analysis altogether. This tightened the focus of the paper without losing too much information and allowed us to streamline our discussion.

We nonetheless thought it worthwhile to respond in detail to each of the reviewer's insightful criticisms. In the following, we use gray font to indicate those responses that have been obviated by our decision to remove Google Trends, but that we nonetheless find useful to report because we find this debate to have scholarly value.

Do the authors have some way to justify their measure (other than that others are using it)?

We use Google Trends not because others are using it, but because others have repeatedly identified Google Trends as a valuable proxy to predict real-life consequences. To give but one example, cross-regional relative volume of searches for a racial slur has been found to be associated with a reduced likelihood of voting for Barack Obama (Stephens-Davidowitz, S. (2014). The cost of racial animus on a black candidate: Evidence using Google search data. Journal of Public Economics, 118, 26-40). We acknowledge that Google Trends must be used with caution.

If the authors can justify the issues in the previous paragraph, then next up is to justify their use of the very few keywords they chose. Do these keywords measure the intent of users in the way that they expect? What way do they expect?

We used the same keywords as for the media coverage. All we are suggesting is that searches for Russia/Mueller reflect the public's interest in that issue. We are not sure what intent users might have when they search for Russia/Mueller other than to inform themselves about Russia/Mueller. This face-validity approach follows recommended practice (Mellon, J. (2014). Internet search data and issue salience: The properties of Google Trends as a measure of issue salience. Journal of Elections, Public Opinion & Parties, 24, 45-72.) and avoids circularity or the detection of spurious correlations that might occur when keywords are chosen from a set after they have been found to correlate with some other measure of interest.

Have they looked at what websites come up (in different parts of the country) when different types of users search for these keywords?

Google Trends does not permit identification of “different types of users”. A regional search would be possible in principle, but in the absence of any theoretical expectations about regional differences, we refrained from conducting this search as it would open the door to unconstrained ad-hoc reasoning.

That is, we need to first identify the quantity of interest in order to evaluate the measure.

We are unsure what is meant by this. We stipulated the quantity of interest (relative search volume for the keywords of interest) a priori and beyond that, with all the necessary caveats, we do not know how else we could “identify” or “evaluate” the measure.

In this case, I would think that the quantity of interest must have to do with either the types of websites that users want to find or the ones they do find by performing a Google search. If so, then the authors could do a study to ascertain whether their measure has good statistical properties. Is the measure (i.e., Google Trends for each keyword or for all together if they like) an unbiased estimator of the quantity of interest?

See above. We assume that people search for Russia/Mueller because they are interested in Russia/Mueller. In the absence of survey data at the same level of granularity this cannot be empirically confirmed or disconfirmed.

And we need to figure out both precision and recall -- or false positives and false negatives. How often does searching for “China” or “jobs” indicate that the user is looking for the same topic Trump sparked in his Tweets? And also, how many are looking for the topic of interest but are missed by the same keywords? To do the latter, the authors would need to identify a large group of keywords representing their theoretical concept.

All of those comments point towards an expansion and foregrounding of the Google Trends analysis. This seems to run counter to the reviewer’s in-principle concerns about Google Trends, which would point to backgrounding the analysis. As already noted, we resolved this tension by dropping Google Trends altogether. In our view, this has strengthened the focus of the paper without incurring a notable loss in content.

Another issue is that the authors need to explain what data were collected, the structure of the data, and basic facts like the unit of analysis, what justifies their timeframe, and how many observations they have -- all before reporting their regression results. Otherwise, there’s no motivation to read the results at all.

We have followed the conventional Nature style which is to have Methods at the end, but we agree that we could have said more up front to make the Results easier to follow. In the revision, we have taken care to provide sufficient information up front to facilitate reading of the Results.

Yes, I realize that methods comes after, but what is the point of me saying that my evaluation of this paper is 197.43 unless you know the scale first?

A fundamental issue is what causal identification assumptions the

authors are claiming. I do not see a discussion of this critical issue. I would think that the results are all afflicted by post-treatment bias, at a minimum, almost by definition (if I'm guessing what the authors are doing correctly). And there would seem to be other issues, but the burden of proof is on the author here and I don't see a case being made.

Indeed, and our extensive response to Reviewer 1 above summarizes the effort we have put into rectifying this problem. (We do not reiterate those details here).

The NY Times is a highly liberal news outlet. I would think the authors should match it with a reasonable conservative outlet, like the WSJ. It would make their results more compelling if they found a similar result.

We justified the choice of The New York Times as the newspaper of record in the paper, based on much previous research. Concerning the alleged liberal bias, a recent quantitative analysis of more than 800,000 news articles through a combination of machine learning and human judgment (involving a sample of nearly 800 judges) has identified the New York Times as being (somewhat) more centrist than The Wall Street Journal (Budak, C., Goel, S., & Rao, J. M. (2016). Fair and balanced? quantifying media bias through crowdsourced content analysis. Public Opinion Quarterly, 80, 250-271, e.g., Figure 1a.).

Of course one could always ask for more analyses but in our view the analysis of the two acknowledged principal agents in print and broadcast media is sufficient for an existence proof of the diversion and suppression we observe. Extending this to other media is a task for future research.

Do the statistical models fit the data? The outcome is a count (the number of diverting keywords), which would normally be analyzed by event count models rather than linear regression.

We have replicated the OLS results using a Negative Binomial Regression (nbreg) model that is designed for counted data. The model yields comparable results and is reported in the online supplement (Table S10). Note that the model cannot be applied to suppression of average coverage because the average no longer represents counted data. The model also did not converge for the suppression model for ABC, which is a common problem with this type of model. Because OLS (and the linear 3SLS) is more readily interpretable as well as being more robust, we report those models in the text.

These data (and models) tend to be heteroskedastic, which would be ignored by linear regression. And probably some of the predicted values of the regression will be negative, which of course makes no sense.

As detailed in the Method, we used robust heteroskedasticity-consistent estimation in the OLS. We have not encountered any negative predicted values (see, e.g., Figure 2) within the observed range of our variables.

Reviewers' comments:

Reviewer #1 (Remarks to the Author):

The authors' responses to my concerns are appropriate and significantly strengthen the credibility of the inferences offered in the article. While I still have remaining concerns, establishing causality in observational time series data is extremely difficult and I now think the manuscript is worth publishing in its current form.

Reviewer #2 (Remarks to the Author):

The authors have responded constructively to all of my suggestions and offered a reasonable response in the several places where they were unable to do so. That leaves me inclined to recommend publication. I still like the study and find it somewhat more persuasive with the revisions implemented by the authors. I particularly like the new Figure 2, which does a nice job of laying out the basic relationships. I find the causal identification issues raised by R1, though certainly real concerns, somewhat less disqualifying than did R1, particularly since I think the authors have done about all one could reasonably be expected to do with observational data to make the case. As I wrote in my original review, extending the traditional agenda setting story to presidential social media behavior is sufficiently novel, in my view, to be interesting even if we don't yet know how generalizable it is.

One minor stylistic comment: Figures 3 and 4 should have more descriptive labeling so that it isn't necessary to read the very long figure captions to understand what the various graphs represent. My general rule of thumb is that if you need to read the caption to understand the figure, something is amiss.

A second minor comment: I'm not sure Figure 1 contributes much. It seems to me that this pattern is easily explained in words. I would prefer to see the authors drop the figure, and then use the extra space to report additional analyses (such as perhaps running a diversion test on "tax", as I suggested below, which appears to be the second-most-frequent term in Table 6).

I do still wonder about the reverse causality question. The authors explored the possibility of reverse causality several ways, one of which is the question of pre-emption: might Trump use tweets to distract from impending Mueller news? They didn't see evidence of this. But I still wonder about that. While I agree with the authors that it is improbable that Trump's tweets on unrelated topics would drive up news about Mueller, I do think that Trump might tweet to

inoculate himself against impending harmful news about Mueller. Trump certainly knew when to expect revelations from the investigation, which frequently appeared on Fridays. And media leaks anticipated many of these revelations. So, he “could” have anticipated some of them at least. For instance, he might tweet about “Fake News” or “The Failing New York Times” on day $t-1$, which might then be followed by a spike in Mueller coverage on day t . Probably this would be a quite different logic than tweeting about “jobs” on day t to depress Mueller news on day $t+1$. The former might be termed inoculation, while the latter would be diversion, as the authors argue. Without the counterfactual we don’t know how big a spike in Mueller coverage there would have been absent the prior tweets. And this may have less to do with agenda setting than with framing the “bad news” from Mueller in a less damaging way (inoculation). But it’s another angle that could be interesting to explore that arguably falls within the broader scope of strategic use of tweets by Trump to shape media coverage of him and his administration.

Lastly, I really like the new Table 6. The frequency of the word “jobs” on this list truly jumps out. It made me think that a word cloud would be instructive and would support some of the choices the authors made (though I don’t think it’s essential since all the information is in the table; it’s just a question of rapid digestion of the data). As noted above, eyeballing the list, I think “tax” might be #2, which, in turn, suggests exploring that is a possible diversion focus.

Reviewer #3 (Remarks to the Author):

NCOMMS-19-31135A

Donald Trump and strategic diversion in the age of Twitter

It is ok that the authors have opinions, but readers should not based scientific conclusions on opinions. At the end of the day, judgements of the authors should not be treated as evidence. For example: “We find it implausible that measurement error plays a significant role in our setting” or “it is logically difficult to see how Trump’s tweets on China, jobs, or immigration could lead to more news about the Mueller investigation on the same day” or “It appears highly implausible to us that all of the variation in Trump’s tweets might be systematically explainable” -- It does not matter whether the authors find this implausible; it matters that they can demonstrate that that the assumption is correct. The absence of evidence which the authors are claiming does not indicate that readers should believe these results. The authors need to find positive evidence for their claims.

“ the fact that we predict news today with Trump's tweets from yesterday (because of the reasons just cited, namely that reporters cannot respond instantaneously) reduces concerns about reverse causality.” -- It is perhaps helpful but it ignores expectations.

“we believe that the sum total of our efforts addresses the omitted variables explanation as much as we possibly can” I think it is reasonable to claim that this is as far as the authors can go with their design, but that does not suggest publication. It suggests the authors tried hard, did all they could, and came up short. They deserve substantial credit for this as scholars, but effort is not what we are judging here. The question is whether scholarly literature should rely on these results as knowledge about the world and to guide future research. For that, we need positive evidence.

3sls indeed is one method capable under certain assumptions of dealing with endogeneity problems, but the assumptions need justification beyond opinions. I'm afraid I do not see that.

I think the placebo tests are better justified than other parts of the paper.

Does the LS analyses fit the data? Running a count model makes sense, but it does not address whether the OLS models, which the authors rely on, fit the data. Also, computer programs (even when they report lack of convergence) should not be making substantive decisions about what model to base conclusions on. Lack of convergence is indicative of collinearity, flat likelihoods, miscoding, lack of model fit, or something else the authors need to identify. The 106 control variables include numerous perhaps reasonable but arbitrary decisions and so we definitely need evidence on fit here. Squared date terms? Lagged variables “as needed”?

The claim “negative binomial regressions are known to suffer from frequent convergence problems” is incorrect; lack of convergence happens for specific reasons, every one of which translates directly into substantive issues the authors should pursue.

“Because OLS (and the linear 3SLS) is more readily interpretable” -- This is incorrect; count models are easy to interpret, it is just that they must be interpreted in different ways. And the difference is not a problem; it is the substantive reason they are to be chosen in the first place with count data. Just compute a quantity of interest directly rather than trying to use the coefficients, which are not of direct interest in most cases.

“we used robust heteroskedasticity-consistent estimation in the OLS” -- Did these alternative SEs make a difference? If so, then the OLS model is misspecified. If not, then why are they being

used here? It is ok to use these as a test of model misspecification by the way.

10 lags for noisy data like these seems surprising. Is this a coding error or typo? If not, then I would pursue this as it may indicate something of substantive interest.

Response to reviewers

(Original text in courier font, replies in *black italics*.)

Reviewers' comments:

Reviewer #1 (Remarks to the Author):

The authors' responses to my concerns are appropriate and significantly strengthen the credibility of the inferences offered in the article. While I still have remaining concerns, establishing causality in observational time series data is extremely difficult and I now think the manuscript is worth publishing in its current form.

We appreciate the referee's response and we are especially grateful for their comments in the first round, as we agree that responding to these has substantially improved the paper.

Reviewer #2 (Remarks to the Author):

The authors have responded constructively to all of my suggestions and offered a reasonable response in the several places where they were unable to do so. That leaves me inclined to recommend publication. I still like the study and find it somewhat more persuasive with the revisions implemented by the authors. I particularly like the new Figure 2, which does a nice job of laying out the basic relationships. I find the causal identification issues raised by R1, though certainly real concerns, somewhat less disqualifying than did R1, particularly since I think the authors have done about all one could reasonably be expected to do with observational data to make the case. As I wrote in my original review, extending the traditional agenda setting story to presidential social media behavior is sufficiently novel, in my view, to be interesting even if we don't yet know how generalizable it is.

One minor stylistic comment: Figures 3 and 4 should have more descriptive labeling so that it isn't necessary to read the very long figure captions to understand what the various graphs represent. My general rule of thumb is that if you need to read the caption to understand the figure, something is amiss.

We have taken this suggestion seriously and we hope that the new layout is more intuitive. For the sake of consistency, we have also added similar labels to the remaining figures.

A second minor comment: I'm not sure Figure 1 contributes much. It seems to me that this pattern is easily explained in words. I would prefer to see the authors drop the figure, and then use the extra space to report additional analyses (such as perhaps running a

diversion test on "tax", as I suggested below, which appears to be the second-most-frequent term in Table 6).

Indeed, we wondered after the first round of revisions whether Figure 1 contributes enough to stay in the paper. We now removed it, following your advice, and adjusted the surrounding discussion.

I do still wonder about the reverse causality question. The authors explored the possibility of reverse causality several ways, one of which is the question of pre-emption: might Trump use tweets to distract from impending Mueller news? They didn't see evidence of this. But I still wonder about that. While I agree with the authors that it is improbable that Trump's tweets on unrelated topics would drive up news about Mueller, I do think that Trump might tweet to inoculate himself against impending harmful news about Mueller. Trump certainly knew when to expect revelations from the investigation, which frequently appeared on Fridays. And media leaks anticipated many of these revelations. So, he "could" have anticipated some of them at least. For instance, he might tweet about "Fake News" or "The Failing New York Times" on day $t-1$, which might then be followed by a spike in Mueller coverage on day t . Probably this would be a quite different logic than tweeting about "jobs" on day t to depress Mueller news on day $t+1$. The former might be termed inoculation, while the latter would be diversion, as the authors argue. Without the counterfactual we don't know how big a spike in Mueller coverage there would have been absent the prior tweets. And this may have less to do with agenda setting than with framing the "bad news" from Mueller in a less damaging way (inoculation). But it's another angle that could be interesting to explore that arguably falls within the broader scope of strategic use of tweets by Trump to shape media coverage of him and his administration.

This is an interesting comment and worthy of exploration. However, we believe that a complete treatment sits outside the scope of the present paper for several reasons, some of them technical and others conceptual.

Concerning the technical issues, we explored the specific suggestion in the preceding comment by checking how many of Trump's tweets contained some of the keywords you mention here. The term "failing New York Times" (case insensitive) appears only 20 times throughout our sample period; the term "failing" appears 62 times. We consider those numbers to be too small to systematically explore inoculation. We cannot rule out that Trump would use other, perhaps similar terms to try and channel media attention away from impeding news that he deems detrimental. However, we have no way of knowing what those would be.

A further complication of this analysis is the conceptualization of what happened before Trump tweets about "fake news" or the "failing New York Times." He usually refers to a particular story when making these allegations, so any empirical model would have to somehow take that into account. This would substantially complicate a regression analysis and introduce further parameters that would be difficult to operationalize, in particular because those parameters may be different for different distracting keywords.

Turning to conceptual issues, the point raised above corresponds to a slightly different research question, namely "inoculation", as opposed to diversion. It is known that if people are warned about the specific techniques by which they might be misled, they then become resilient to misinformation (e.g., Cook, J., Lewandowsky, S., & Ecker. U. K. H. (2017). Neutralizing Misinformation Through Inoculation: Exposing Misleading Argumentation Techniques Reduces Their Influence. PLOS

ONE, 12, e0175799). On that basis, one might indeed expect Donald Trump to tweet diversions ahead of time, assuming he had prior knowledge of impending media coverage. However, this is a different line of rhetoric altogether that requires a separate research project.

Nonetheless, following the reviewer's suggestion we now mention this potential line of exploration in the Discussion. In particular, we now highlight the paper by Ross and Rivers (2018) that has already taken a first step in that direction.

Lastly, I really like the new Table 6. The frequency of the word "jobs" on this list truly jumps out. It made me think that a word cloud would be instructive and would support some of the choices the authors made (though I don't think it's essential since all the information is in the table; it's just a question of rapid digestion of the data). As noted above, eyeballing the list, I think "tax" might be #2, which, in turn, suggests exploring that is a possible diversion focus.

A word cloud is a neat idea and we now show a cloud in the new Figure 3. We find the word cloud visually appealing and impactful. We also retained the table because it provides the precise numbers for the corresponding information and it permits identification of the most relevant word combinations.

With respect to analyzing "tax" as an additional distractor, we have two responses. First, this was not our starting hypothesis, whereas "China", "jobs", and "immigration" was. Thus, it would be unprincipled to incorporate that keyword in our first set of analysis (the "targeted" analysis), because the identification of "tax" emerged during the expanded analysis.

We nonetheless checked whether the occurrence of "tax" in Trump's tweets would produce interesting insights. What we find is a positive and statistically significant uptick following NYT news on the Mueller/Russia investigation – but the coefficient associated with the ABC news is not statistically significant (yet still positive). When it comes to the second leg of our analysis (whether "tax" tweets predict less Mueller/Russia news tomorrow), we do not find statistically significant coefficients, neither for the NYT nor for the ABC. Thus, the empirical evidence for "tax" as a diversionary term remains weaker than for the main ones we investigate here.

In sum, we hope these responses sufficiently address Reviewer 2's comments and remaining concerns. Once again, we are grateful for the suggestions we received in the process of preparing this paper.

Reviewer #3 (Remarks to the Author):

It is ok that the authors have opinions, but readers should not based scientific conclusions on opinions. At the end of the day, judgements of the authors should not be treated as evidence.

First, it must be noted that when Reviewer 3 is referring to "readers" in this context, s/he is referring to the few readers of the "response-to-reviewers" document from our previous submission: All of the quotations by Reviewer 3 below are taken from our response to the reviewers rather than our paper (although one is replicated in the paper). Responses to reviewers—in this case all were replies to Reviewer 1—are conventionally stated in the context of specific criticisms and serve a very different purpose from the article itself. We are unaware of any precedent in which one reviewer critiques a paper not by considering the manuscript itself, but a covering letter written in response to another reviewer (who accepted our arguments).

Second, we must clarify that our statements are not “opinions” but rather aim to illustrate what we can and cannot test with the data, in the context of the criticisms offered by Reviewer 1 at the previous round. Endogeneity concerns (i.e., reverse causality, omitted variables, and/or measurement error) can only be alleviated by careful statistical and conceptual analyses that ultimately involve judgment. All empirical work in social sciences—or indeed any other science— involves a certain amount of judgment (as opposed to “opinions”). The best a researcher can do is to provide as many tests as possible to support their judgment. We believe that is what we did in this paper and in the response to reviewers. Notably, Reviewer 1 agreed with our judgments, as indicated by their positive response at this round, notwithstanding their critical stance at the first round.

For example: “We find it implausible that measurement error plays a significant role in our setting”

A researcher can never fully exclude the possibility that measurement error is present— indeed, if you could measure it then it wouldn’t be a problem in the first place. This is not only true for observational data in the social sciences but for all measurements anywhere. It therefore inevitably becomes a matter of judgment—based on reasoning, theory, observation, or a mix thereof—whether measurement error might play a significant role.

Our judgment that measurement error was an unlikely explanation for the results was embedded in multiple supporting arguments in the response-to-reviewers document. To reiterate, we capture all NYT articles and ABC news segments as available in the corresponding archives, and there is no reason to expect omissions in these archives to bias our results—to do so would imply that archiving errors are sensitive to Mueller/Russia coverage, and in our judgment we do not see how this could occur with algorithmic scraping. Similarly, we study all of Trump’s tweets during the sampling period and we are not aware of Twitter’s archive being affected by random or indeed systematic errors. The remaining margin for measurement error would therefore only extend to Trump misspelling words (e.g., writing “immigration” as “imigration”). Our keyword search would indeed miss the corresponding mention in that case. In our judgment, however, it is implausible that misspellings play a significant role in our setting.

or “it is logically difficult to see how Trump’s tweets on China, jobs, or immigration could lead to more news about the Mueller investigation on the same day”

As we noted in the paper and the response to reviewers, the news cycle necessarily includes a lag: journalists have to process the information, they must write articles, they are vetted by editors and fact-checkers, and then (even for online reporting) they must be typeset before they can be published. All of this takes time. It follows that a tweet by Donald Trump (unless issued early in a day) is unlikely to elicit any increased media coverage on the same day. Moreover, even setting aside those obvious technical considerations, we are unable to contrive a logical mechanism by which a Trump tweet on China, jobs, or immigration could systematically lead to more news about the Mueller investigation on the same day. There may be such a mechanism, but the onus is on critics to explain this mechanism if they disagree with our judgment. Note that we emphasized “systematically” in our claim because for this to be an issue, such (as yet unknown and temporally unlikely) dynamics would have to operate not only a couple of times throughout our sampling period, but frequently enough to produce a systematic bias in our analysis. That is highly unlikely; but we cannot fully exclude that possibility, of course, which is why our language was cautious.

or “It appears highly implausible to us that all of the variation in Trump’s tweets might be systematically explainable” -- It does not matter whether the authors find this implausible; it matters that they can demonstrate that that the assumption is correct.

Our claim refers to the theoretical possibility that all of Trump's tweets (i.e., all words used that we capture in our content analysis) are explainable by variables that we could consistently measure in our sample. It is impossible to disprove this theoretical possibility. However, an examination of its implications highlights its implausibility. For example, what measurable causal variable would explain why Donald Trump misspells describing (as "discribing") in a tweet that also mistakes an apostrophe for a hyphen? (<https://twitter.com/realDonaldTrump/status/1177539052683309056?s=20>). What variable would explain the use of the words "LameStream Media" in the same tweet? Any claim that all variation in Trump's tweets might be systematically captured would have to postulate a perfect explanation of even such arbitrary choices. That is highly implausible. Indeed, we would be concerned if anyone ever found it plausible that all variation in any social science dataset could be explained without over-fitting.

The absence of evidence which the authors are claiming does not indicate that readers should believe these results. The authors need to find positive evidence for their claims.

To summarize: First, all statements mentioned by Reviewer 3 in the foregoing come from our response to Reviewer 1 regarding endogeneity concerns. They were surrounded by the underlying logic that would motivate these statements, so they should be considered in context. The third statement, which is replicated in the paper, is followed by a detailed explanation why it is conceptually difficult to hypothesize a systematic effect of Trump tweets about China, jobs, and immigration on NYT/ABC news coverage of the Mueller/Russia investigation on the same day.

We have, however, taken the reviewer's concern seriously and have reworded the explanation in the text to express even more intellectual humility.

Second, none of our claims cited by Reviewer 3 are subject to adjudication by evidence. There is no "positive evidence" that could be adduced for or against them. All of our claims are however supported by judgment. We stand by our judgments. We emphasize that Reviewer 3 does not take issue with our particular judgments, but the very fact that we made them. This criticism is a priori inescapable because judgment is a necessary ingredient of any scientific work.

"the fact that we predict news today with Trump's tweets from yesterday (because of the reasons just cited, namely that reporters cannot respond instantaneously) reduces concerns about reverse causality." -- It is perhaps helpful but it ignores expectations.

Because the reviewer does not articulate these expectations, we are unsure how these unknown expectations could explain our second set of results (Trump's tweets on China, jobs, and immigration predict significantly less news coverage of the Mueller/Russia investigation).

It bears repeating that it was a specific expectation, articulated in the paper, that motivated our hypothesis – namely that Trump would expect coverage of the Mueller/Russia investigation to continue and he would try and distract from that coverage with tweets on China, jobs, and/or immigration (which represent his political strengths).

"we believe that the sum total of our efforts addresses the omitted variables explanation as much as we possibly can" I think it is reasonable to claim that this is as far as the authors can go with their design, but that does not suggest publication. It suggests the authors tried hard, did all they could, and came up short.

Reviewer 1, who was very critical at the first round, did not share this judgment. Their response to our efforts at this round was: "The authors' responses to my concerns are appropriate and significantly strengthen the credibility of the inferences offered in the article. While I still have

remaining concerns, establishing causality in observational time series data is extremely difficult and I now think the manuscript is worth publishing in its current form."

They deserve substantial credit for this as scholars, but effort is not what we are judging here. The question is whether scholarly literature should rely on these results as knowledge about the world and to guide future research. For that, we need positive evidence.

We must beg to differ. No matter how much "positive evidence" one provides, ultimately it is impossible to fully resolve all endogeneity concerns with any kind of data. To our knowledge, no single paper has ever been published that has put to rest endogeneity concerns with absolute certainty. Our statements interpreting the results reflect the necessary degree of scientific humility.

3sls indeed is one method capable under certain assumptions of dealing with endogeneity problems, but the assumptions need justification beyond opinions. I'm afraid I do not see that.

We are not sure what the referee means here. The main assumption of our joint regression analysis is that the second part of our analysis (whether tweets lead to less news coverage of the Mueller/Russia investigation) is conceptually built on the first. Thus, a joint estimation is conceptually appropriate. We have not seen the use of 3SLS justified in the literature beyond stating the fact that applies here—namely that we have a coupled system of relationships, and joint modeling is a superior way of analyzing those coupled systems.

This method has been used throughout a host of papers to date (e.g., see the seminal papers on R&D spillovers by Audretsch and Feldman, American Economic Review 1996, and Ashenfelter and Rouse, Quarterly Journal of Economics 1998; cited 7,104 and 819 times, respectively), especially when the dependent variables exhibit path dependency. This is indeed the case in our setting (see Tables S1-S3).

I think the placebo tests are better justified than other parts of the paper.

Does the LS analyses fit the data? Running a count model makes sense, but it does not address whether the OLS models, which the authors rely on, fit the data.

We report fit (R^2) in Table 2.

Also, computer programs (even when they report lack of convergence) should not be making substantive decisions about what model to base conclusions on. Lack of convergence is indicative of collinearity, flat likelihoods, miscoding, lack of model fit, or something else the authors need to identify. The 106 control variables include numerous perhaps reasonable but arbitrary decisions and so we definitely need evidence on fit here. Squared date terms?

The selection of control variables was not arbitrary. All of them are based on sound theoretical analysis of the variables that may influence Trump's tweets and, in turn, may influence NYT/ABC coverage of the Mueller/Russia investigation. For example, the squared date term aims to control for nonlinear time trends in Trump's tweets (first hypothesis) and news coverage of the

Mueller/Russia investigation (second hypothesis). Theoretically, it is possible that those variables did not follow a linear trend over time, and that is what the squared term is trying to capture. Nevertheless, our results are qualitatively unchanged when excluding that variable. In addition, we include a fixed effect for each week of the sampling period to capture week-to-week fluctuation of the variables under consideration that are not already captured by the time trends. This accounts for a host of different political events that could independently affect both NYT/ABC reporting and Trump's tweeting behavior, such as events involving North Korea or Iran to name just a couple. There is nothing arbitrary about controlling for variables that are known to exert an effect on political events—namely time (as a proxy for world events) at various levels of granularity.

Lagged variables "as needed"?

Lagged variables "as needed" refers to how far back we may expect autocorrelation. We provide detailed evidence for that in the supplementary part of the paper (Tables S1-S3). Thus, the control variables pertaining to lagged variables follow empirical evidence on autocorrelation. We agree with Reviewer 3 that our phrasing in the paper ("as needed to account for autocorrelations") did not do this careful examination justice. We have replaced this by "as suggested by a detailed examination of the underlying autocorrelation structures (see Tables S1-S3)." We apologize for the lack of clarity.

The claim "negative binomial regressions are known to suffer from frequent convergence problems" is incorrect; lack of convergence happens for specific reasons, every one of which translates directly into substantive issues the authors should pursue.

Negative binomial regression (nbreg) models can be problematic in converging when some control variables are soaking up "too much" statistical variation, which can especially be the case for fixed effects that are motivated by underlying theory. However, in practice, results from OLS models and nbreg-type models usually produce very similar conclusions. For example, we refer to "Regression Analysis of Count Data" (Cameron & Trivedi, 2013).¹ In such cases, it has become common practice to use OLS models instead, particularly if the results from less restrictive nbreg models are virtually identical to the corresponding OLS results. This is the case here.

In general, researchers often opt for OLS models because they allow including variables whose inclusion is conceptually warranted but that a more restrictive nbreg model would struggle with due to collinearity issues. This is partly why we decided to make the linear models our main ones. Nevertheless, it must be restated that our findings are consistent when using an nbreg model.

"Because OLS (and the linear 3SLS) is more readily interpretable" -- This is incorrect; count models are easy to interpret, it is just that they must be interpreted in different ways.

We are unaware of the existence of a 3SLS count model. As noted earlier, the 3SLS model is a superior approach to analysis of coupled systems of equations, and this would simply not be possible with a count model.

¹ To illustrate, Cameron & Trivedi (2013) write in Section 3.7.1: "Nonetheless, OLS estimates in practice give results qualitatively similar to those for Poisson and other estimators using the exponential mean... the most highly statistically significant regressors from OLS regressions, using OLS output t statistics, are in practice the most highly significant using Poisson regressions."

And the difference is not a problem; it is the substantive reason they are to be chosen in the first place with count data. Just compute a quantity of interest directly rather than trying to use the coefficients, which are not of direct interest in most cases.

We are not sure we understand. Count models have coefficients, just like any other regression model, and those coefficients require interpretation (see, e.g., Table S10 in the supplement). We therefore do not know how we could “compute a quantity of interest directly rather than trying to use the coefficients”.

“we used robust heteroskedasticity-consistent estimation in the OLS”
-- Did these alternative SEs make a difference?

The supplementary material contains tables that compare different type of standard errors. None of the conclusions are materially affected by the choice of how to calculate standard errors..

If so, then the OLS model is misspecified.

To the extent that we understand this point, we must beg to differ. Those standard errors were developed precisely to cope with situations such as heteroskedasticity—indeed, why would they exist if they did not have a remedial effect? This has been well established in the statistical literature; see, e.g. Wooldridge’s Introductory Econometrics: A Modern Approach (2016), going back to White (Econometrica, 1980).

If not, then why are they being used here?

Because we conform to long-standing statistical practice.

It is ok to use these as a test of model misspecification by the way.

10 lags for noisy data like these seems surprising. Is this a coding error or typo?

No, this is neither a coding error nor a typo. The corresponding regressions are available in Table S1. We are also unsure what is thought to be surprising about that number. If the data on Trump tweets were more noisy (taken here to mean white noise), we would expect less autocorrelation, if anything (because the noise drowns out the autocorrelation signal).

To assuage any fears about this particular number (10), we checked and found all our results to be virtually identical when we expand the set of lagged variables to 21 (even though there is no evidence for autocorrelation of that length, which is why we did not report that analysis in the first place).

If not, then I would pursue this as it may indicate something of substantive interest.

We are not sure what it is that would be of substantive interest. However, given that the results are virtually identical with lagged variables to 21, we are fairly confident that we have not missed anything interesting.

REVIEWER COMMENTS

Reviewer #2 (Remarks to the Author):

The authors have done a good job responding to my several remaining concerns and suggestions. I am generally satisfied at this point and think the paper is ready for publication. It is a very nice piece of research and I look forward to seeing it published.

Reviewer #3 (Remarks to the Author):

Manuscript # NCOMMS-19-31135B

Title Donald Trump and strategic diversion in the age of Twitter

“We assumed that the president would divert attention from Mueller to topics that he perceived to represent his particular strengths. We considered the three keywords “China”, “jobs”, and “immigration” as playing to those strengths....” -- Here, the authors need to demonstrate that these three keywords select Tweets with the characteristics described. The claim seems plausible, but there needs to be evidence provided rather than assumed. At some times (like now), referring to any one of these words could select Tweets that play to his weaknesses.

The authors introduce Table 1 and Figure 1 without explaining what they are, what the unit of analysis is, what the measures are, and precisely how to interpret the results (in the text anyway). No specific quantity of interest is identified or estimated other than various regression coefficients, which may be related but are different.

More importantly, fundamental causal claims are made without identifying restrictions being explained, claimed, or justified. The main claimed contribution of the paper would seem to depend on this, and all that is provided is a set of associations -- interesting associations, but causality requires clear identifying assumptions.

Figure 1 includes confidence intervals that do not come close to proper statistical coverage. This implies that the model does not fit the data and needs fixing.

The Oster test is interesting, but the problem is not (or not only) omitted variables. The problem is endogeneity and the lack of clear identifying assumptions. The paper includes no identification strategy. An enormous literature has arisen over the last several decades precisely characterizing what these are and developing many statistical methods that are able to at least provide clues about causal effects and, when one can defend the specific mathematical assumptions, can directly estimate them.

I like the placebo analysis, although it is very specific to the cases chosen. Can the authors give evidence that it was fairly chosen rather than cherry picked, even inadvertently?

Methods:

No justification is given for the specific keywords chosen except that it seems plausible to the authors. The history of automated text analysis indicates that humans are not good at selecting keywords to classify social media posts. If the methods in this paper are different, the authors need to justify that. I do not see even an attempt to do so.

Various rules text analysis rules are introduced. These may be plausible, but the result needs to be justified. Do these rules select the Tweets of interest? This does not seem to be addressed.

If the negative binomial model is not converging, it suggest a problem with the specification, not a technical problem with computer code.

The univariate time series analyses of each variable separately do not have much relation to the multivariate analysis.

Overall, this analysis choice seems to be extremely model dependent, using an approach to data analysis that is dated and difficult to justify.

Reviewer #4 (Remarks to the Author):

This paper is about two posited causal relationships. First, that Trump responds to coverage of Russia and the Mueller investigation by producing “diversionary” tweets. Second, that these tweets then have the intended consequences: they reduce subsequent Russia/Mueller coverage by the media.

I agree with other reviewers and the authors that causal inference is difficult in this setting. Clearly, the causal interpretation is of substantial interest and is given frequently. For example, Figure 1 talks about “effects of media coverage of Russia/Mueller on diversionary tweets”. The authors should explicitly state the assumptions required for the causal interpretation of these results, so that readers can readily evaluate their plausibility.

The study of media coverage is focused on only two outlets. This certainly has consequences for the precision and generality of measurement of any effects of media choices. But perhaps this is fine.

Sensitivity analysis

I appreciated the use of sensitivity analysis allowing for some unobserved confounding. This is something more observational papers should have. I think the discussion of this could be made clearer.

In particular, I think what the authors say about δ is not correct. In particular:

“Values of $\delta > 1$ are taken to support the robustness of a model because it suggests that the observables are at least as important as the unknown hidden variables” (p. 13).

Rather, according to Oster (2019) larger δ s correspond to the the unobservables needing to be more strongly related to treatment than the observables in order to explain away the observed association. So the question is whether there are other unobservables that are causing the timing of the Mueller coverage to this higher degree, which may also be causes of Trump’s tweeting.

Furthermore, even that interpretation is somewhat misleading. I might suggest referring to Cinelli & Hazlett (2020, section 6.3), which also discuss some confusions with using the results in Oster (2019). In particular, they note that δ can only be given the interpretation I wrote above under some more restrictive assumptions about the relationship of the confounders with the outcome.

The authors did not say much about why this approach is not taken for the other relationship of interest, “diversionary” tweets on subsequent coverage (i.e. Table 2 only has results for columns 1-3 of Table 1, not columns 4-6). My guess is that it is much more sensitive to unobserved confounding, given the relationship is much less strong (and not particularly statistically strong as noted below).

Anyway, I think it is nonetheless good that the authors did this sensitivity analysis, but they should be more accurate in how they describe what it means. I also am not clear about how precisely, e.g., δ is estimated given all the autocorrelation.

Choice of diversionary keywords

The authors write that “we conducted a targeted analysis in which the diversionary topics were stipulated a priori to be those that play to President Trump’s political strengths, based on our analysis of his political position during the first two years of this presidency.” (p. 6) Given the quite general stated motivation for the choice of these keywords, I think readers can reasonably wonder about how these were picked and whether they were picked without any knowledge of the observed events being analyzed.

Statistical inference

This is a challenging setting for statistical inference, as there are really just two time series of interest, and they presumably involve a good amount of autocorrelation. The main results use Newey–West SEs allowing for some autocorrelation. I have not scrutinized the choices here; but I hope the authors have thought carefully about whether these are credible given the length of their panel and the number of lags (e.g., are the asymptotic approximations good enough?).

Some of the results seem to be more clearly distinguishable from the null than others. In particular, the association between Mueller coverage and “diversionary” tweets by Trump is highly statistically

significant. On the other hand, the association of his tweeting with subsequent coverage is not so clear. I believe (using the rounded estimate and SE in Table 1) that the relevant p-value is around 0.023. So, while significant at the 0.05 threshold, this is not particularly strong statistical evidence. (Here I am using the results that combine the NYT and ABC, which are the strongest statistically.) We might worry that, say, if the asymptotic approximation is not so good, then these SEs are somewhat too small and this result is less clear.

Placebo tests

The authors test some other null hypotheses using the same method that they expect to be true. This is a common practice, though it can also be somewhat misleading, because just because a placebo quantity isn't statistically significant doesn't mean it is statistically significantly different from the quantity of interest (Gelman & Stern 2006). The authors could take an equivalence test approach here (Hartman & Hidalgo 2018). But actually I think what they have done is reasonably compelling and could be regarded as an something like a null distribution to which the observed estimates could be compared (and are compared, somewhat informally).

Other comments:

- I'm not sure why the authors cite their ref. 32 (Rohner & Frey) in support of rather than Merrill, J. (1999). The global elite: World's best newspapers reflect political changes. IPI Report 1999, 13– 15, which seems to be the source for the choices in Rohner & Frey, except for some broad, speculative argumentation by Rohner & Frey.

- Side note: I have not previously seen the method from Oster (2019) referred to solely as "the Oster test". I think the literature typically characterizes the approach in Oster (2019) as a further development in a long line of work, including prior developments by, e.g., Rosenbaum and Imbens.

- "Given the lag time of news reports, there is little opportunity for a tweet to generate coverage within the same 24-hour period." (p. 12) Is this true? Doesn't the nightly news usually cover what has happened that day?

Summary

I think this is an interesting topic for study. There seems to be some reasonable evidence for the idea that Mueller coverage proceeds and perhaps causes Trump to tweet about other topics. The evidence for the next link — that this works by distracting the media — seems to have much less strong evidence. So I think the authors should be careful in their claims about the latter and the editorial team should consider the relative importance and contribution of new evidence for each claim.

Sincerely,
Dean Eckles

Cinelli, C., & Hazlett, C. (2020). Making sense of sensitivity: Extending omitted variable bias. *Journal of*

the Royal Statistical Society: Series B (Statistical Methodology), 82(1), 39-67.

Gelman, A., & Stern, H. (2006). The difference between “significant” and “not significant” is not itself statistically significant. *The American Statistician*, 60(4), 328-331.

Hartman, E., & Hidalgo, F. D. (2018). An equivalence approach to balance and placebo tests. *American Journal of Political Science*, 62(4), 1000-1013.

Reviewers' comments:**Reviewer #2 (Remarks to the Author):**

The authors have done a good job responding to my several remaining concerns and suggestions. I am generally satisfied at this point and think the paper is ready for publication. It is a very nice piece of research and I look forward to seeing it published.

We thank the reviewer for their helpful and constructive comments throughout, which have improved the paper considerably. We are delighted that Reviewer 2 is now endorsing publication.

Reviewer #3 (Remarks to the Author):

"We assumed that the president would divert attention from Mueller to topics that he perceived to represent his particular strengths. We considered the three keywords "China", "jobs", and "immigration" as playing to those strengths...." -- Here, the authors need to demonstrate that these three keywords select Tweets with the characteristics described. The claim seems plausible, but there needs to be evidence provided rather than assumed. At some times (like now), referring to any one of these words could select Tweets that play to his weaknesses.

This is a good point that is related to one raised by Reviewer 4. We respond to this point in our reply to Reviewer 4 at length below.

We agree that right now, during the pandemic, tweeting about jobs is unlikely to be the president's preferred topic—however, we did not choose the keywords now, but at a time when they were clearly pointing to the president's preferred topics, as we show extensively in the revision.

The authors introduce Table 1 and Figure 1 without explaining what they are, what the unit of analysis is, what the measures are, and precisely how to interpret the results (in the text anyway).

Following Nature convention, those details were provided in the Methods. However, we agree that this can sometimes make it hard to follow the details of the Results, and so we have now added several additional pointers to the Methods wherever necessary.

No specific quantity of interest is identified or estimated other than various regression coefficients, which may be related but are different.

We have sought to clarify this further. In particular, we re-introduced a figure from the first submission that graphically represents the hypothesized relationships within our regression framework.

More importantly, fundamental causal claims are made without identifying restrictions being explained, claimed, or justified. The main claimed contribution of the paper would seem to depend on this, and all that is provided is a set of associations -- interesting associations, but causality requires clear identifying assumptions.

We have revised the relevant discussion, further toning down claims about causality and/or explaining identifying restrictions. The expanded discussion has particularly benefitted from Reviewer 2's initial and Reviewer 4's additional suggestions, as detailed below.

Figure 1 includes confidence intervals that do not come close to proper statistical coverage. This implies that the model does not fit the data and needs fixing.

We are unsure what the reviewer is referring to. The confidence envelopes in the figure (now Figure 2) are graphically representing significant effects of the predictors (which are reported in Table 1). We are at a loss to see what requires "fixing".

The Oster test is interesting, but the problem is not (or not only) omitted variables. The problem is endogeneity and the lack of clear identifying assumptions. The paper includes no identification strategy. An enormous literature has arisen over the last several decades precisely characterizing what these are and developing many statistical methods that are able to at least provide clues about causal effects and, when one can defend the specific mathematical assumptions, can directly estimate them.

As we state on pp. 15-21 (Lines 208-310), omitted variables are one of three manifestations of endogeneity, and we now discuss all three. (In the previous version we did not discuss measurement error because, as we now state explicitly, it is highly unlikely to play a role here given the nature of our measurements—but yes, for completeness this can also be mentioned).

To summarize, we perform the following steps to support identification, which in the absence of opportunity for controlled experimentation is the most extensive check we can perform:

- *A careful exploration of autocorrelation concerns.*
- *A discussion about why reverse causality is unlikely to be a major concern (note: it is impossible to fully resolve these theoretical problems empirically with observational data).*
- *Sensitivity analyses to quantify the likelihood that omitted variables are able to explain away our findings.*
- *3SLS analyses to jointly estimate our two hypotheses, using two different temporal windows for the suppression effect.*
- *Placebo analyses to check whether unobservable variation that is potentially common to other contemporary topics (e.g., Brexit) can produce comparable findings.*

Remarkably, the results of all those explorations are consistent with our hypotheses. Nevertheless, we are aware that it is impossible to fully resolve endogeneity concerns, and the language throughout the paper now reflects that sentiment.

I like the placebo analysis, although it is very specific to the cases chosen. Can the authors give evidence that it was fairly chosen rather than cherry picked, even inadvertently?

This is a good point. We justified the Brexit analysis at length because Brexit resembles Russia/Mueller in many important respects with the crucial exception of not being a political threat to the president (p. 22). As far as the other items are concerned, they were chosen to cover a broad range of topics that are unlikely to be politically challenging at all (gardening, skiing, football, vegetarian) or are unlikely to challenge the president (economy, during our sampling period which predated COVID). The topics were chosen without knowledge of the data, to cover a broad range of possible arenas—from politics to sports to cooking to hobbies. We could have chosen other topics, of

course, as there is a nearly infinite number of plausible topics that are covered by the New York Times. We cannot rule out that some topics may yield different effects—although we did not explore any other topics—but the uniformity of results across all placebo topics makes that less likely.

Methods:

No justification is given for the specific keywords chosen except that it seems plausible to the authors. The history of automated text analysis indicates that humans are not good at selecting keywords to classify social media posts. If the methods in this paper are different, the authors need to justify that. I do not see even an attempt to do so.

Various rules text analysis rules are introduced. These may be plausible, but the result needs to be justified. Do these rules select the Tweets of interest? This does not seem to be addressed.

This is a very good point. We have addressed this by doing a full-vocabulary analysis on the chosen tweets and also all the various NYT articles that were selected through our keywords. We report the results of this analysis as word clouds which, in our view, compellingly demonstrate the adequacy of our method. Specifically, Figure 1 shows the semantic space for Trump’s diversionary tweets, and Figures 4 and 8 summarize the content of NYT articles that were selected for the various placebo analyses.

If the negative binomial model is not converging, it suggest a problem with the specification, not a technical problem with computer code.

Our response to this is twofold: First, for the average-media model, which combines all available data and is therefore most powerful, the negative binomial model is not appropriate. This is because the combined data do not represent counts but rather z-scores. Use of OLS for those data is entirely appropriate. It follows that the convergence issue is only relevant for a subset of the analyses.

Second, it is well known that negative binomial regression (nbreg) models can be problematic in converging when some control variables are soaking up a lot of statistical variation, which is particularly likely for fixed effects that are motivated by underlying theory. In those cases, one interpretation (a statistical one) is that the corresponding variable should not be in the model. However, conceptually, that is not a great answer if the researchers have grounds to believe that the variable does matter in providing potential explanatory power for the outcome variable.

For example, if a researcher runs cross-country regressions and believes that unobservables at the country level could plausibly influence both the outcome variable and the main independent variable of interest, then country-fixed effects provide an important remedy. However, it is often the case that sample data are not sizeable enough to ‘accommodate’ all country-fixed effects in a more restrictive estimation framework, such as nbregs. Thus, based on statistical considerations alone, one may be tempted to ignore country-fixed effects. However, if there is reason to believe such fixed effects do matter, then another, often preferred option is to switch to a less demanding model, most often OLS, that would allow incorporating country-fixed effects.

What is commonly done then is to present negative binomial models where possible and compare the derived coefficients with those from the respective OLS regressions that include the same set of regressors. Now, if the derived results are closely comparable in terms of sign and statistical significance, as they are in our case, then a next step would be to proceed with the more basic OLS model that does allow for the inclusion of additional controls (e.g., country-fixed effects).

We are not aware of another strategy that would allow the researcher to draw as much information from the underlying data as possible. Our approach is well recognized in the literature.

For example, we refer to “Regression Analysis of Count Data” (Cameron & Trivedi, 2013).¹ This book has been cited more than 8,700 times.

In summary, the convergence issue has no bearing on the most powerful analysis. To the extent that it arises, there is widespread precedent that OLS is a suitable substitute in such cases, in particular when the two approaches give rise to the same results (as they do in our case; see Supplement).

The univariate time series analyses of each variable separately do not have much relation to the multivariate analysis.

We are not sure what the reviewer is referring to. If the reviewer is referring to the 3SLS modelling, then we do not understand this claim because comparison of Tables 1 and 3 shows how the separate OLS analyses relate to the 3SLS, and the figures summarizing the expanded analysis allow the same comparison between panels. In all cases, the 3SLS analysis confirms the pattern of the independent OLS models.

Overall, this analysis choice seems to be extremely model dependent, using an approach to data analysis that is dated and difficult to justify.

In the absence of any specific suggestion about what analysis would be less dated and better justified, it is difficult to respond to this critique. Our analysis is justified and relies on techniques published in 2019-2020 (e.g., Cinelli & Hazlett, 2020; see reply to Reviewer 4 below), as well as commonly accepted workhorse models in empirical analyses.

Further, the main results are confirmed across all kinds of alternative models now (OLS, 3SLS, etc.); that is, our results remain robust, consistent, and independent of the specific empirical model we apply.

Reviewer #4 (Remarks to the Author):

This paper is about two posited causal relationships. First, that Trump responds to coverage of Russia and the Mueller investigation by producing “diversionary” tweets. Second, that these tweets then have the intended consequences: they reduce subsequent Russia/Mueller coverage by the media.

I agree with other reviewers and the authors that causal inference is difficult in this setting. Clearly, the causal interpretation is of substantial interest and is given frequently. For example, Figure 1 talks about “effects of media coverage of Russia/Mueller on diversionary tweets”. The authors should explicitly state the assumptions required for the causal interpretation of these results, so that readers can readily evaluate their plausibility.

We share the concern about the difficulties we are facing. We have refrained from using words such as “effect” that imply causality. We have expanded and further caveated our discussion

¹ To illustrate, Cameron & Trivedi (2013) write in Section 3.7.1: “Nonetheless, OLS estimates in practice give results qualitatively similar to those for Poisson and other estimators using the exponential mean... the most highly statistically significant regressors from OLS regressions, using OLS output t statistics, are in practice the most highly significant using Poisson regressions.”

of the requirements for causality (pp.15-21).

The study of media coverage is focused on only two outlets. This certainly has consequences for the precision and generality of measurement of any effects of media choices. But perhaps this is fine.

There are of course many other outlets we could have focused on, but a choice had to be made due to obvious practical constraints. We believe we made a principled choice and selected the most appropriate outlets. We nonetheless acknowledge this limitation in the Discussion on p. 35.

Sensitivity analysis

I appreciated the use of sensitivity analysis allowing for some unobserved confounding. This is something more observational papers should have. I think the discussion of this could be made clearer. In particular, I think what the authors say about δ is not correct. In particular:

“Values of $\delta > 1$ are taken to support the robustness of a model because it suggests that the observables are at least as important as the unknown hidden variables” (p. 13).

Rather, according to Oster (2019) larger δ s correspond to the the unobservables needing to be more strongly related to treatment than the observables in order to explain away the observed association. So the question is whether there are other unobservables that are causing the timing of the Mueller coverage to this higher degree, which may also be causes of Trump’s tweeting.

See below; this is no longer pertinent because we have replaced the Oster test with the new methodology suggested by the reviewer. We are grateful for this discussion, as we do believe the new tests have made the paper stronger.

Furthermore, even that interpretation is somewhat misleading. I might suggest referring to Cinelli & Hazlett (2020, section 6.3), which also discuss some confusions with using the results in Oster (2019). In particular, they note that δ can only be given the interpretation I wrote above under some more restrictive assumptions about the relationship of the confounders with the outcome.

We thank the reviewer for drawing our attention to this recent article. Having read the work by Cinelli and Hazlett (C&H), we decided to discard the Oster test in favour of their new approach. We found C&H’s methodology to be more easily interpretable and, as the reviewer pointed out, it compares favourably to the Oster test in terms of robustness (e.g., sensitivity to degrees of freedom, C&H, p. 63).

The authors did not say much about why this approach is not taken for the other relationship of interest, “diversionary” tweets on subsequent coverage (i.e. Table 2 only has results for columns 1-3 of Table 1, not columns 4-6). My guess is that it is much more sensitive to unobserved confounding, given the relationship is much less strong (and not particularly statistically strong as noted below).

This point has been addressed by reporting the new analysis for both diversion and suppression. As the reviewer surmised, the suppression result is established with less statistical precision than the result pertaining to diversion. Nonetheless, the results of the sensitivity analysis

are encouraging for the suppression model as well (in our judgment; see text for details). The sensitivity analysis also underscores the strength and robustness of the diversion model.

Anyway, I think it is nonetheless good that the authors did this sensitivity analysis, but they should be more accurate in how they describe what it means. I also am not clear about how precisely, e.g., δ is estimated given all the autocorrelation.

All our analyses were informed by a detailed exploration of the pattern of autocorrelations, which is reported in the online supplement. We included the necessary lags in all analyses. This is now additionally highlighted in the text early on (whereas previously it was only mentioned in the Methods). We believe that this should address the reviewer's concern about autocorrelations.

Choice of diversionary keywords

The authors write that "we conducted a targeted analysis in which the diversionary topics were stipulated a priori to be those that play to President Trump's political strengths, based on our analysis of his political position during the first two years of this presidency." (p. 6) Given the quite general stated motivation for the choice of these keywords, I think readers can reasonably wonder about how these were picked and whether they were picked without any knowledge of the observed events being analyzed.

This is a good point. We did not preregister the method because we knew it would evolve during the project, and hence the a priori nature of our choice is difficult to establish. Although the methods were not preregistered, keywords were picked without any knowledge of the observed events being analysed.

We also did not use a formal algorithm to establish Trump's political strengths (because no such algorithm exists), and instead relied on our analysis of media coverage, pundit commentary, and Trump's campaign topics. We are citing several additional sources, including a Politifact analysis of Trump's campaign and also his own re-election campaign website (see p. 12, Lines 170-180). Politifact is a reputable independent fact checker owned by the non-profit Poynter Centre. We are not aware of a more formal process that we could have used.

We now also report a full-vocabulary analysis of the diversionary tweets (word cloud in Figure 1) which provides the full context of the diversion. We believe that anyone familiar with American politics will recognize Trump's preferred campaign issues in those tweets.

Moreover, we believe that none of those issues matter greatly in light of the expanded analysis, which used all available tweet topics. Use of all available tweets prevents any possibility of "cherry-picking" of keywords. The expanded analysis therefore underscores the robustness of our targeted analysis. The fact that the expanded analysis by and large identified the same keywords that were used in the targeted analysis (word cloud in Figure 6 [formerly Figure 3]) provides further support for our choice of keywords.

In addition, we also decided to change our terminology slightly: instead of talking about the president's "strengths" we now talk about his "preferred topics." Whereas talking about "strengths" requires a political imputation, the fact that our keywords refer to Trump's "preferred topics" can be established more unambiguously.

Statistical inference

This is a challenging setting for statistical inference, as there are really just two time series of interest, and they presumably involve a good amount of autocorrelation.

We have considerable understanding of the autocorrelations because they were explored in depth in Tables S1 through S4 in the Supplement. We have now highlighted this further in the main body of the text.

The main results use Newey–West SEs allowing for some autocorrelation. I have not scrutinized the choices here; but I hope the authors have thought carefully about whether these are credible given the length of their panel and the number of lags (e.g., are the asymptotic approximations good enough?).

As noted above, we have checked the autocorrelations carefully and the analyses in the Online Supplement should allay those concerns. We apologize if our previous version did not make this clear enough. We now added references to the corresponding analysis of autocorrelations on p. 12.

Some of the results seem to be more clearly distinguishable from the null than others. In particular, the association between Mueller coverage and “diversionary” tweets by Trump is highly statistically significant. On the other hand, the association of his tweeting with subsequent coverage is not so clear. I believe (using the rounded estimate and SE in Table 1) that the relevant p-value is around 0.023. So, while significant at the 0.05 threshold, this is not particularly strong statistical evidence. (Here I am using the results that combine the NYT and ABC, which are the strongest statistically.) We might worry that, say, if the asymptotic approximation is not so good, then these SEs are somewhat too small and this result is less clear.

This is a fair point and if we had relied on the targeted analysis alone, our suppression result may not appear to be terribly strong. However, we would argue that the expanded analysis deals with that because it shows that the relationship between tweeting and subsequent coverage generalizes across a fairly large cluster of keywords and across media outlets (bottom-right quadrants the last three panels in Figure 5 [formerly Figure 2]). Nonetheless, we acknowledge repeatedly now that the evidence for suppression is not as strong as that for diversion.

Placebo tests

The authors test some other null hypotheses using the same method that they expect to be true. This is a common practice, though it can also be somewhat misleading, because just because a placebo quantity isn’t statistically significant doesn’t mean it is statistically significantly different from the quantity of interest (Gelman & Stern 2006). The authors could take an equivalence test approach here (Hartman & Hidalgo 2018). But actually I think what they have done is reasonably compelling and could be regarded as an something like a null distribution to which the observed estimates could be compared (and are compared, somewhat informally).

We are aware of the problem raised by Gelman and Stern, and we have addressed this in several ways. We acknowledge that the reviewer’s comments have some force for the targeted analysis, and we have dealt with this by combining the Russia-Mueller and Brexit models into a single system of SUR (Seemingly Unrelated Regression) equations, and then added the constraint of setting one or the other coefficient (for Brexit or Russia-Mueller) to zero. This leads to significant loss of fit for Russia-Mueller but not Brexit: this result goes beyond the previous analysis because both models are now part of a single system of equations. (Note that constraining the two coefficients to be equal

does not incur a loss of fit, presumably because of the highly imprecise estimate of the Brexit coefficient; see new Footnote b).

In addition, we believe that the sum total of our placebo analyses (i.e., across the 8 panels in Figures 7 and 8 [formerly Figure 4]) provide more than an informal comparison to a null distribution. The visual and numeric contrasts between Figures 7 and 8 on the one hand, and Figure 5 on the other, seem sufficiently striking, in particular given that each panel in all figures contains a formalized null distribution.

Recall that the contour lines in each panel show the distribution that would be expected if there were no association between the variables under consideration, obtained by randomizing the Twitter timeline. Accordingly, in Figures 7 and 8 relatively few points fall outside these contours, whereas Figure 5 is replete with points in the bottom-right quadrant.

Other comments:

- I'm not sure why the authors cite their ref. 32 (Rohner & Frey) in support of rather than Merrill, J. (1999). The global elite: World's best newspapers reflect political changes. IPI Report 1999, 13- 15, which seems to be the source for the choices in Rohner & Frey, except for some broad, speculative argumentation by Rohner & Frey.

We have replaced the citation as suggested.

[REDACTED]

[REDACTED]

- Side note: I have not previously seen the method from Oster (2019) referred to solely as "the Oster test". I think the literature typically characterizes the approach in Oster (2019) as a further development in a long line of work, including prior developments by, e.g., Rosenbaum and Imbens.

This is a good point but no longer relevant because we no longer report the Oster test results.

- "Given the lag time of news reports, there is little opportunity for a tweet to generate coverage within the same 24-hour period." (p. 12) Is this true? Doesn't the nightly news usually cover what has happened that day?

We have revised this claim to focus on the fact that there is less opportunity for a tweet to lead to same-day coverage than for the reverse to occur. We believe that this is less controversial: whereas a tweet can be formulated and broadcast in seconds or minutes, media coverage even in the digital era takes at least hours to create. Thus, for example, any tweet issued in the afternoon is unlikely to be reported on the ABC evening news that day whereas Trump may well tweet immediately after the evening news have been shown. It is also unlikely that the NYT would be quick enough to respond within hours although we have not been able to quantify the lag time.

Summary

I think this is an interesting topic for study. There seems to be some reasonable evidence for the idea that Mueller coverage proceeds

and perhaps causes Trump to tweet about other topics. The evidence for the next link – that this works by distracting the media – seems to have much less strong evidence. So I think the authors should be careful in their claims about the latter and the editorial team should consider the relative importance and contribution of new evidence for each claim.

We have gone through the paper carefully and have further caveated our conclusions, noting in particular the difference in the strength of evidence for the diversion (strong) and suppression (less strong) models.

REVIEWERS' COMMENTS

Reviewer #4 (Remarks to the Author):

I think the authors have made some improvements to the paper and generally reasonably represent the remaining uncertainty about the causal relationships of interest, including some notes that the evidence that Trump "successfully" uses distraction is more limited than the evidence that he responds with his "preferred topics" (I like this new terminology better as well). And nice to see the new sensitivity analysis.

I still found the Conclusion a bit sweeping on this point:

"it is clear that President Trump is a master at exploiting this tool" (line 486). This goes beyond the present quantitative evidence I think, even if we are fully convinced about the successful distraction point, this seems like a pretty basic strategy that presumably many politicians deploy (and might even be better at?). If the authors want to say something like this, I might suggest stating that many observers have remarked something like that.

One small suggestion is that the authors might want to replace "Twitter" in the title with something more general. Describing this as the "age of Twitter" is a bit funny given Twitter still has a much smaller number of active users (in the US and elsewhere) than competitors. And presumably this would work similarly if Trump was using Instagram instead.

Sincerely,
Dean Eckles

Reviewer's comments:

Reviewer #4 (Remarks to the Author):

I think the authors have made some improvements to the paper and generally reasonably represent the remaining uncertainty about the causal relationships of interest, including some notes that the evidence that Trump "successfully" uses distraction is more limited than the evidence that he responds with his "preferred topics" (I like this new terminology better as well). And nice to see the new sensitivity analysis.

I still found the Conclusion a bit sweeping on this point: "it is clear that President Trump is a master at exploiting this tool" (line 486). This goes beyond the present quantitative evidence I think, even if we are fully convinced about the successful distraction point, this seems like a pretty basic strategy that presumably many politicians deploy (and might even be better at?). If the authors want to say something like this, I might suggest stating that many observers have remarked something like that.

We have rephrased this to read: "It is clear that President Trump exploits this tool, although it remains to be seen whether Trump's skills in that regard are unique to him or will be picked up by future presidents and other leaders around the globe."

One small suggestion is that the authors might want to replace "Twitter" in the title with something more general. Describing this as the "age of Twitter" is a bit funny given Twitter still has a much smaller number of active users (in the US and elsewhere) than competitors. And presumably this would work similarly if Trump was using Instagram instead.

We agree about the broader point, but we also believe that Donald Trump is widely known to rely on Twitter to conduct much official and personal communication. To express those two ideas, we have retitled the manuscript Donald Trump's tweets and political diversion in the age of social media.